JCB Journal of Cell Biology

# Phosphatidylserine synthesis controls oncogenic B cell receptor signaling in B cell lymphoma

Jumpei Omi[1], Taiga Kato[2], Yohei Yoshihama[2], Koki Sawada[1], Nozomu Kono[1], and Junken Aoki[1]

**Cancer cells harness lipid metabolism to promote their own survival. We screened 47 cancer cell lines for survival dependency on phosphatidylserine (PS) synthesis using a PS synthase 1 (PTDSS1) inhibitor and found that B cell lymphoma is highly dependent on PS. Inhibition of PTDSS1 in B cell lymphoma cells caused a reduction of PS and phosphatidylethanolamine levels and an increase of phosphoinositide levels. The resulting imbalance of the membrane phospholipidome lowered the activation threshold for B cell receptor (BCR), a B cell–specific survival mechanism. BCR hyperactivation led to aberrant elevation of downstream $Ca^{2+}$ signaling and subsequent apoptotic cell death. In a mouse xenograft model, PTDSS1 inhibition efficiently suppressed tumor growth and prolonged survival. Our findings suggest that PS synthesis may be a critical vulnerability of malignant B cell lymphomas that can be targeted pharmacologically.**

## Introduction

Metabolic reprogramming is a hallmark of cancer cells and uses distinct modes of energy metabolism to support massive growth, metastasis, and immune evasion (Faubert et al., 2020). Cancer cells also employ sophisticated changes in phospholipid metabolism to achieve efficient survival and expansion (Cheng et al., 2016). The phospholipid composition of biological membranes, especially the plasma membrane (PM), is an important determinant of signal transduction via various receptors, which is exploited by cancer cells to promote oncogenic survival signals (Bi et al., 2019). Cancer-related changes in the phospholipidome have recently been targeted in antitumor drug development (Bunney and Katan, 2010).

An emerging target in cancer therapy is phosphatidylserine (PS; Sekar et al., 2022; Yoshihama et al., 2022), an anionic phospholipid that has serine as a head group and constitutes 2–15% of cellular phospholipids (Vance, 2018). In mammals, PS synthase 1 (PTDSS1) and PTDSS2 synthesize PS by directly incorporating serine into the head group of phosphatidylcholine (PC) and phosphatidylethanolamine (PE), respectively (Kuge et al., 1991, 1997). This process, referred to as a base-exchange reaction, takes place at contact sites between the endoplasmic reticulum (ER) and mitochondria (Stone and Vance, 2000). The newly synthesized PS is transported to the PM, where it is highly enriched in the cytoplasmic leaflet. This asymmetric distribution of PS shapes its classical functions as a scaffold for blood coagulation on the surface of activated platelets (Wang et al., 2022) and as an "eat-me" signal on apoptotic cells, which expose PS on the extracellular leaflet to be captured by PS

receptors on phagocytotic cells (Segawa and Nagata, 2015). PS is important in cancer immunology because cancer cells expose a high level of PS on the outer PM leaflet under non-apoptotic conditions to suppress immune cell activity (Birge et al., 2016). This transmembrane "flip-flop"-based function for cell-to-cell communication is not the whole story of PS, however. Previous reports revealed that deficient or excess PS synthesis significantly affects the metabolism and intracellular transport of other phospholipid classes inside cells (Chung et al., 2015; Sohn et al., 2016; Vance, 2008).

In addition to being a precursor of PS, PE is a product of PS decarboxylation catalyzed by PS decarboxylase (PISD), an enzyme that localizes exclusively in the mitochondria (Vance, 2008). Mammalian cells that are deficient in PTDSS1-catalyzed PS synthesis show a concomitant decrease in PE in the absence of an exogenous supply of PS (Kuge et al., 1986). Furthermore, impairment of the PISD reaction results in mitochondrial dysfunction and embryonic lethality in mice (Steenbergen et al., 2005; Tasseva et al., 2013). PS synthesis is also coupled with phosphoinositides (PIPs) metabolism. Pioneering studies revealed that the oxysterol-binding protein-related proteins (ORPs) ORP5 and ORP8 mediate the interorganelle exchange of PS and PIPs at membrane contact sites, whereby phosphatidylinositol (PI)-4-monophosphate (PI4P) or PI-4,5-bisphosphate $(PI(4,5)P_2)$ synthesized in the PM is exchanged with PS synthesized in the ER (Chung et al., 2015; Ghai et al., 2017). This machinery may underlie the cellular pathogenicity of Lenz-Majewski syndrome, in which overproduction of PS by a

---

[1]Department of Health Chemistry, Graduate School of Pharmaceutical Sciences, The University of Tokyo, Tokyo, Japan;   [2]Daiichi Sankyo Co., Ltd., Tokyo, Japan.

Correspondence to Junken Aoki: jaoki@mol.f.u-tokyo.ac.jp.



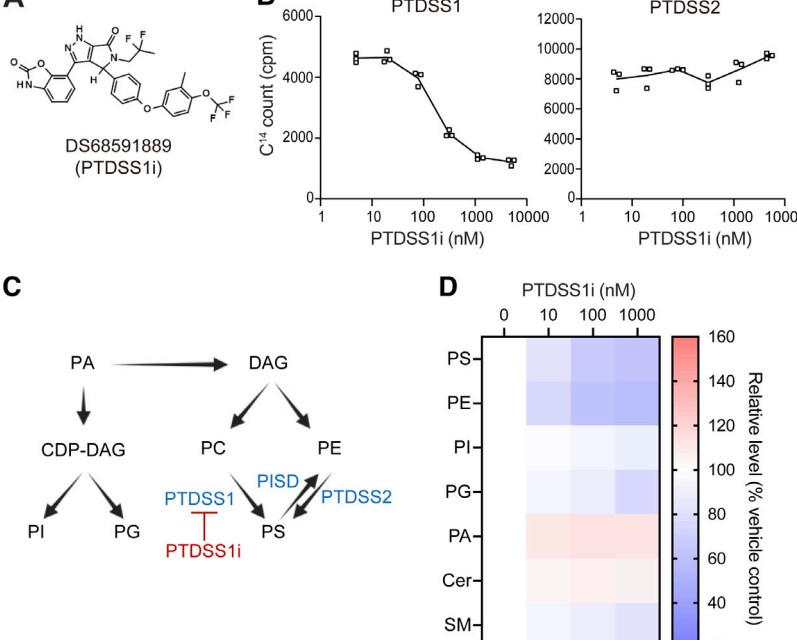

Figure 1. **The structure and activity of DS68591889.** **(A)** Chemical structure of DS68591889 (PTDSS1i). **(B)** The inhibitory activity of PTDSS1i against PTDSS1 (left panel) or PTDSS2 (right panel). The membrane fraction of Sf9 expressing human PTDSS1 or PTDSS2 was incubated with L-[$^{14}$C]-serine and the indicated concentrations of PTDSS1i at 37°C for 20 min. After washing, the scintillation counts (counts per minute, cpm) of the membrane fractions were measured and used for the estimation of PS synthase activity. Data are presented as a dot plot of values for three independent experiments. **(C)** The flow of major glycerophospholipid metabolism. DS68591889 (PTDSS1i) specifically inhibits PTDSS1, which catalyzes serine incorporation into PC. **(D)** The effects of PTDSS1i on the phospholipid compositions of HeLa cells cultured in the presence of the indicated concentrations of PTDSS1i for 2 d. Data are presented as a heatmap of the percentage of the control value without PTDSS1i (mean of three independent experiments).

gain-of-function PTDSS1 variant reduces the level of PIPs in the PM (Sohn et al., 2016; Sousa et al., 2014). Furthermore, recent reports revealed that other ORPs also participate in PS–PI4P countertransport at distinct organelle contact sites (Kawasaki et al., 2021; Mochizuki et al., 2022). PS synthesis thus greatly impacts the membrane phospholipidome. However, its significance, especially in cancer cell biology, has not been elucidated.

We screened a panel of human cancer cell lines for survival dependency on PS synthesis using a recently developed PTDSS1 inhibitor. The results revealed B cell lymphoma as a cancer type whose survivability heavily relies on PS synthesis. Mechanistic analysis revealed that B cell receptor (BCR), a B cell lymphoma-specific machinery, is a critical determinant of the dependency on PS synthesis. An imbalanced phospholipidome in the absence of sufficient PS synthesis causes aberrant and spontaneous BCR activation, leading to cell death. Our findings clarify a novel and targetable vulnerability of phospholipid metabolism in B cell lymphoma.

## Results

### Cell line screens showed that B cell lymphoma is highly dependent on PS

A PTDSS1-specific inhibitor was recently developed based on synthetic lethal activity against cancer cells harboring a genetic deletion of PTDSS2, in which PTDSS1 inhibition causes profound loss of cellular PS and acute cell death (Yoshihama et al., 2022). DS68591889, referred to hereafter as PTDSS1i, is a close analog of a previously reported compound, DS55980254 (Fig. 1 A). PTDSS1i had inhibitory activity against human PTDSS1, but not PTDSS2, in a cell-free PTDSS assay (Fig. 1, B and C). In HeLa cells, PTDSS1i caused a substantial loss of PS and PE (Fig. 1 D), along with a slight decrease in phosphatidylglycerol (PG) and sphingomyelin (SM) and an increase in phosphatidic acid (PA).

We used PTDSS1i to screen a panel of 47 human cancer cell lines for growth dependency on PS synthesis. The results confirmed that PTDSS1i caused a reduction of PS levels in a wide range of cell lines (Fig. 2 A). The PTDSS1 inhibition reduced the levels of major PS acyl-chain species such as C36:1-PS and C34:1-PS, whereas it slightly increased the level of C40:6-PS, which is mainly produced by PTDSS2 (Kimura and Kim, 2013; Fig. 1 B). PTDSS1 inhibition also caused a profound reduction of C36:1-PE and C34:1-PE, indicating substantial PE production from PS species by PS decarboxylase in mitochondria. The levels of C38:4-PI also increased in several cell lines under PTDSS1 inhibition. The levels of PG and PA were also affected by PTDSS1 inhibition but to a lesser extent than those of PS, PE, and PI. These data suggest that inhibition of PTDSS1 by PTDSS1i substantially induces the phospholipid imbalance in a wide range of cancer cells.

The growth of malignant B cell lymphoma-derived lines was strongly suppressed by PTDSS1i (Fig. 2 C). Cell lines derived from Burkitt's lymphoma (Ramos), mantle cell lymphoma (Jeko-1), germinal center B-cell-like diffuse large B cell lymphoma (SU-DHL-6), and activated B-cell-like diffuse large B cell lymphoma (SU-DHL-2) were highly susceptible to PTDSS1 inhibition, whereas lines derived from acute lymphoblastic leukemia (RS4;11 and NALM-6) and multiple myeloma (MM.1S), as well as most solid-tumor lines, were less sensitive.

To further examine the cell type-specific response to PTDSS1i, we generated PS synthase-knockout (KO) clones of the Ramos, SU-DHL-6, HeLa, and A549 cell lines (Fig. S1 A). In these cell lines, PTDSS1 KO did not upregulate PTDSS2 expression and vice versa. In Ramos cells, PTDSS2 KO did not significantly alter the phospholipid composition, whereas PTDSS1 KO resulted in a significant phospholipid imbalance (Fig. 3 A), mainly due to decreases in the dominant PE and PS acyl-chain species (C34:1, C36:1, C36:2) and an increase in C38:4-PI (Fig. 3 B and Fig. S3 B).

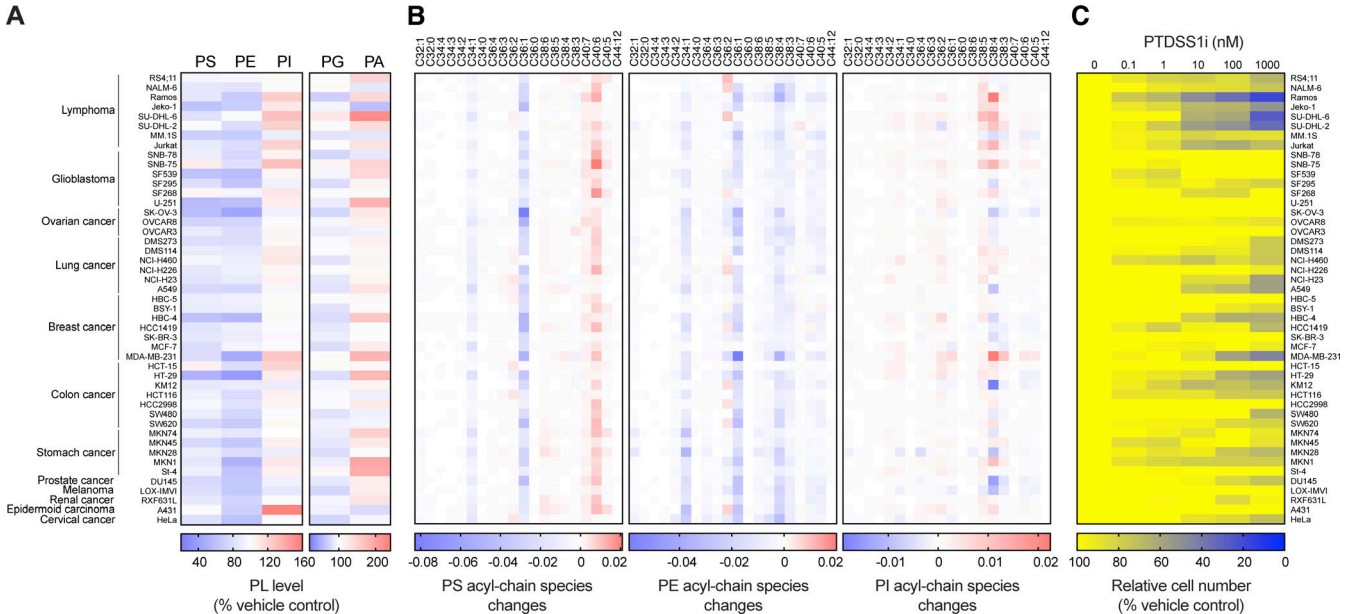

Figure 2. **A screen of cancer cell lines revealed that B cell lymphoma is highly susceptible to PTDSS1 inhibition. (A)** The effects of PTDSS1i on the phospholipid compositions of various cancer cell lines. Data are presented as a heatmap of the percentage of vehicle control. **(B)** The effects of PTDSS1i on the acyl-chain species of PS (left panel), PE (middle panel), and PI (right panel) in various cancer cell lines. Data are presented as a heatmap for the area ratio of PTDSS1i-treated cells subtracted by those of vehicle control. **(C)** The effects of PTDSS1i on the growth of various cancer cell lines cultured in the presence of the indicated concentrations of PTDSS1i for 4–6 d. Data are presented as a heatmap of the percentage of the vehicle control value (mean of two independent experiments).

PTDSS1 KO induced a similar phospholipid imbalance in SU-DHL-6 lymphoma cells (Fig. S3 C). In HeLa and A549 cells, PTDSS1 KO similarly reduced the PS and PE acyl-chain species, but did not affect PI levels (Fig. S3, D and E). The changes in four

cell lines are consistent with those induced by PTDSS1i (Fig. 2 A and Fig. S1, B–E).

Consistent with their relative susceptibilities to PTDSS1i, PTDSS1-KO Ramos clones exhibited more severe growth defects

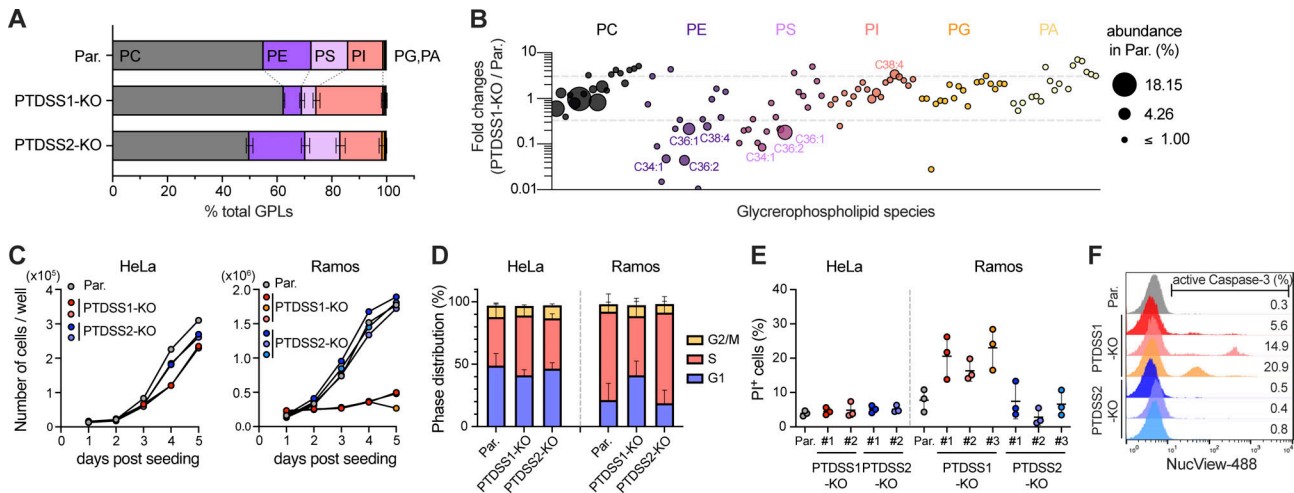

Figure 3. **Phospholipid imbalance, growth retardation, and cell death in PTDSS1-deficient B cell lymphoma. (A)** Major phospholipid composition of parental (Par.), PTDSS1-KO, and PTDSS2-KO Ramos cells. **(B)** Fold changes in acyl-chain species of glycerophospholipid in PTDSS1-KO Ramos cells compared with parental cells. Circle size indicates the abundance of each of 120 phospholipids detected in parental cells. The major phospholipid species (>2% abundance in parental cells) with threefold changes between parental and PTDSS1-KO cells are labeled. Data are presented as the mean of three clones. **(C)** Growth curves of HeLa and Ramos mutants compared with the parental strains. Data are expressed as absolute cell numbers within wells. Representative data from three independent experiments are shown. **(D)** Cell cycle distribution of HeLa and Ramos mutants compared with the parental strains (mean + SD of three independent experiments). **(E)** Accumulation of PI+ HeLa and Ramos cell mutants during culture. Data are presented as the mean ± SD of three independent experiments with a dot plot of values for the individual experiments. **(F)** Activation of caspase-3 in Ramos cells. Percentages of active caspase-3-positive cells are shown.

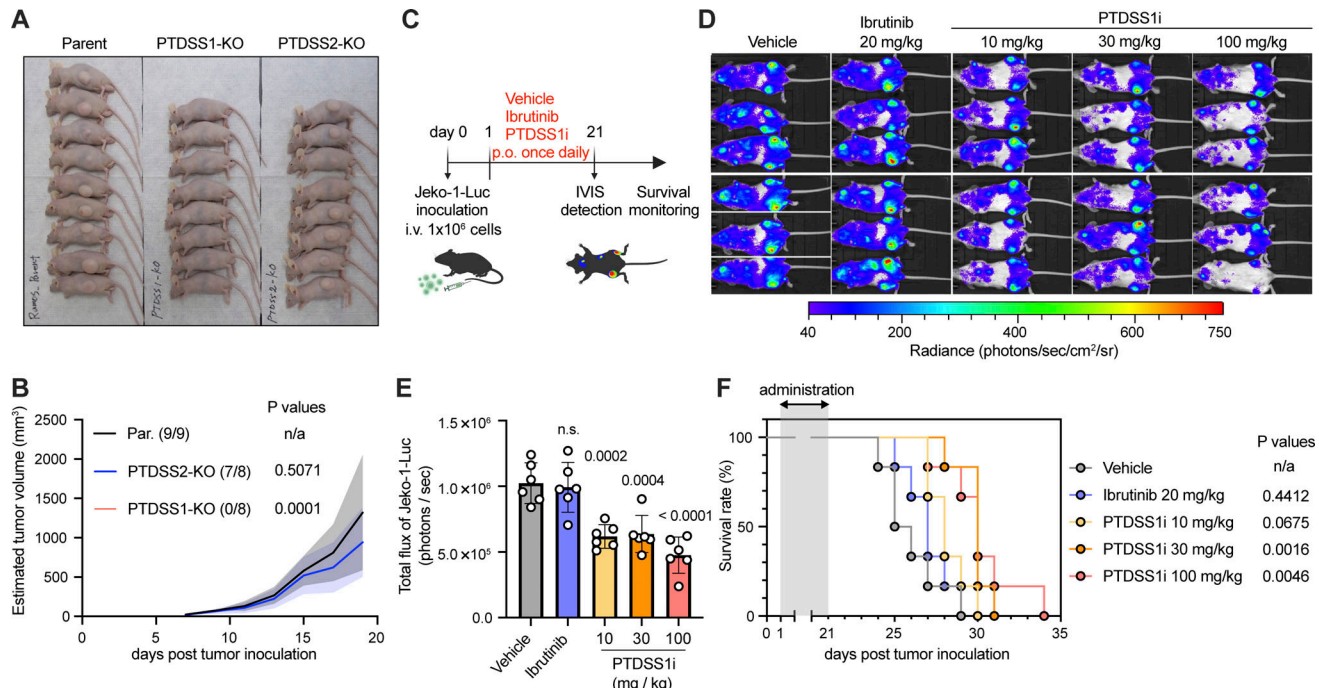

Figure 4. **PS synthesis by PTDSS1 is required for the survival of B cell lymphoma in vivo. (A)** Photograph of mice 19 d after subcutaneous injection with parental (left), PTDSS1-KO (middle), or PTDSS2-KO Ramos clones (right). Visible tumors formed at the left flank of mice injected with parental or PTDSS2-KO clones, but not PTDSS1-KO clones. **(B)** Growth curve of subcutaneous Ramos tumors. Data are presented as the mean ± SD of each group. Fractions in parentheses indicate the number of mice showing tumor engraftment in each group. Note that the curve for PTDSS1-KO overlaps with the x-axis. Statistical P values based on ANOVA followed by one-sided Dunnett's test are shown. n.a., not applicable. **(C)** Schematic of the Jeko-1-Luc intravenous xenograft model. **(D)** In vivo imaging of engrafted Jeko-1-Luc on day 21 after tumor injection. **(E)** Bioluminescence activity (Total flux) of engrafted Jeko-1-Luc cells. Statistical P values based on ANOVA followed by one-sided Dunnett's test are shown. n.s., not significant. **(F)** Survival of Jeko-1-injected mice that received vehicle or the indicated doses of each compound. Statistical P values based on log-rank test are shown.

than PTDSS1-KO HeLa clones (Fig. 3 C), mainly due to growth arrest in the G1 phase (Fig. 3 D) and prominent cell death during culture (Fig. 3 E). PTDSS1-KO SU-DHL-6 lymphoma, but not A549 lung carcinoma, also showed significant cell death (Fig. S1 F). Furthermore, PTDSS1-KO Ramos clones were positive for activated caspase-3 staining (Fig. 3 F), confirming that the cell death was induced by active apoptosis. Taken together, these findings indicate that B cell lymphoma is more heavily dependent on PS synthesis for survival than other cancer types.

## PS synthesis by PTDSS1 is required for the growth of B cell lymphoma in vivo

We next examined the importance of PS synthesis for the growth of B cell lymphoma in vivo. In a subcutaneous xenograft model using BALB/cAJcl-nu (nu/nu) mice, all mice injected with parental Ramos cells developed visible tumors (Fig. 4, A and B). Strikingly, mice injected with a mixture of three PTDSS1-KO clones completely failed to form tumors, whereas mice injected with PTDSS2-KO clones developed tumors comparable to those in mice injected with parental cells. We also tested the in vivo efficacy of another PTDSS1i, DS55980254, in an intravenous xenograft model using Jeko-1 lymphoma cells expressing luciferase (Fig. 4 C). In vivo bioluminescence detection showed that oral treatment with 10, 30, or 100 mg/kg PTDSS1i suppressed Jeko-1 cell engraftment in the bone marrow of mice with systemic spread (Fig. 4, D and E). Consistent with this, treatment

with 30 or 100 mg/kg PTDSS1i for 21 d after Jeko-1 inoculation prolonged survival to a greater degree than did treatment with 20 mg/kg Ibrutinib, a first-in-class Bruton's tyrosine kinase (BTK) inhibitor (Fig. 4 F). These results indicate that PS synthesis by PTDSS1 is essential for the growth of B cell lymphoma in vivo.

## PTDSS1 negatively regulates BCR-induced $Ca^{2+}$ signaling and cell death

PTDSS1i inhibition did not overtly alter the number, position, and morphology of the cellular organelles to exert cytotoxicity in Ramos cells (Fig. S2, A–F). The best-characterized machinery specific to B cells is the BCR complex (Skalet et al., 2005; Yan et al., 2008). It consists of a membrane-bound immunoglobulin (mIg) associated with a CD79A/CD79B heterodimer in the PM (Tolar and Pierce, 2022). Upon ligation with antigens, BCR-associated B cell linker protein (BLNK) and tyrosine kinases such as SYK, BTK, and Lyn form a signalosome that rapidly activates downstream phospholipase Cγ2 (PLCγ2)-mediated $IP_3$ production, leading to $Ca^{2+}$ release from the ER (Young and Staudt, 2013). $IP_3$-mediated $Ca^{2+}$ depletion in the ER induces store-operated $Ca^{2+}$ entry (SOCE), which leads to significant extracellular $Ca^{2+}$ influx to the cytosol via STIM1/2-Orai complexes at PM–ER contact sites (Baba and Kurosaki, 2011). The biological outcomes of BCR-induced $Ca^{2+}$ signaling are highly context-specific. In the presence of sufficient survival signals,

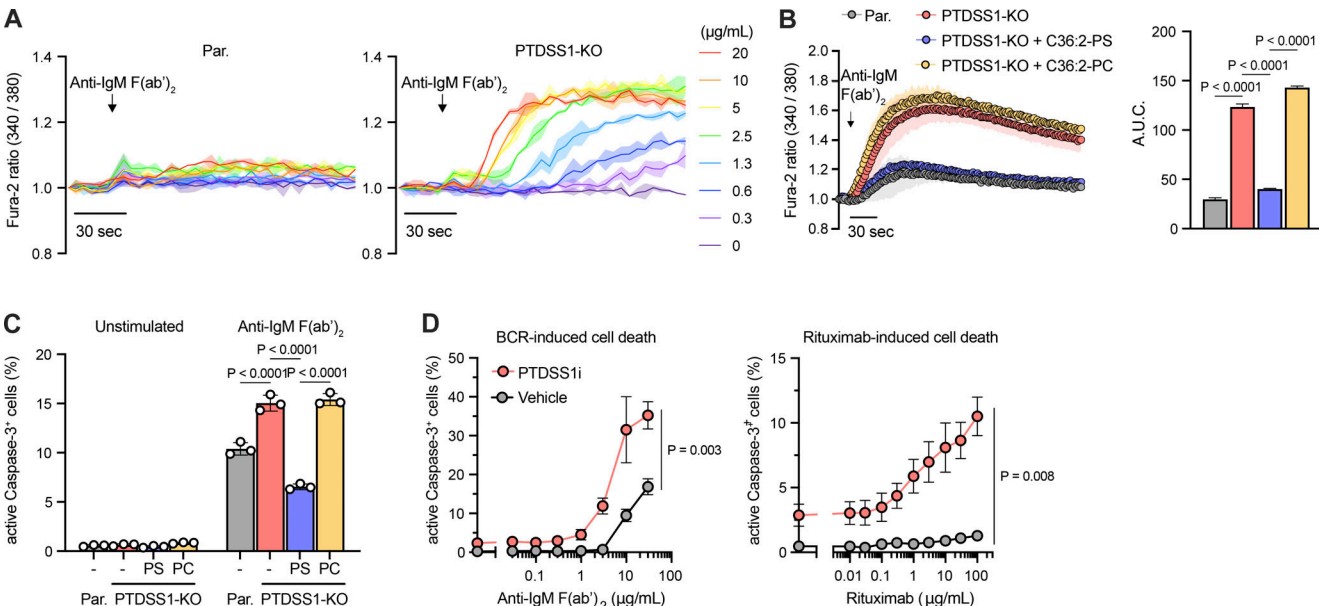

**Figure 5. PTDSS1 negatively regulates BCR-induced Ca²⁺ signaling and cell death. (A)** Ca²⁺ responses of Ramos cells to BCR ligation in the absence of extracellular Ca²⁺. Parental or PTDSS1-KO Ramos cells were stained with Fura-2 and stimulated with the indicated concentrations of anti-IgM F(ab')₂. Representative data are presented as a Fura-2 ratio normalized to the initial value (mean ± SD of two independent measurements). **(B)** Ca²⁺ responses of Ramos cells to BCR ligation in the presence of extracellular Ca²⁺. Ramos cells pretreated with 60 µM 36:2-PS or 36:2-PC were stimulated with 20 µg/ml anti-IgM F(ab')₂ (left panel). The values for area under the curve (A.U.C.) are shown (right panel). Data are presented as the mean ± SD of three independent experiments. Statistical P values based on one-way ANOVA followed by Šídák's multiple comparisons test are shown. **(C)** BCR ligation-induced apoptosis of Ramos cells. Ramos cells pretreated with 60 µM 36:2-PS or 36:2-PC were stimulated with 20 µg/ml anti-IgM F(ab')₂ for 24 h. The NucView488-positive (aCaspase-3⁺) population was then analyzed by flowcytometry. Data are presented as the mean ± SD of three independent experiments. Statistical P values based on one-way ANOVA followed by Šídák's multiple comparisons test are shown. **(D)** Apoptotic response of Ramos cells to BCR ligation (left panel) or rituximab treatment (right panel). Ramos cells cultured under the 100 nM PTDSS1i for 3 d were stimulated with the indicated concentrations of anti-IgM F(ab')₂ (for 24 h) or rituximab (for 48 h), and then the NucView488-positive (aCaspase-3⁺) population was analyzed by flowcytometry. Data are presented as the mean ± SD of three independent experiments.

BCR signals promote cell survival and proliferation (Huntington et al., 2006; Mackus et al., 2002); otherwise, they promote apoptosis via the ER stress responses, which is classically recognized as machinery for negative selection against newly developed B cells in bone marrow (Sandel and Monroe, 1999).

To determine whether PTDSS1 inhibition affects BCR signaling, we monitored Ca²⁺ release from the ER in response to anti-human IgM F(ab')₂ stimulation using a Fura-2 ratiometric Ca²⁺ probe. Whereas wild-type Ramos cells showed only modest cytosolic Ca²⁺ elevation, a PTDSS1-KO clone showed an extremely enhanced response (Fig. 5 A). The PTDSS1-KO clone also showed enhanced SOCE after BCR ligation (Fig. S2 G). These effects could be counteracted by supplying excess amounts of C36:2-PS but not C36:2-PC to the culture medium (Fig. 5 B), confirming the on-target effects of PTDSS1 KO. Consistent with the elevated Ca²⁺ signaling, BCR-mediated apoptosis, as determined by active caspase-3, was also elevated in the PTDSS1-KO clone and PTDSS1i-treated cells (Fig. 5, C and D). Inhibition by PTDSS1i also augmented the apoptotic response to anti-CD20 antibody (rituximab), a widely used therapeutic for B cell lymphoma that kills lymphoma cells by activating intracellular Ca²⁺ signaling (Janas et al., 2005). These findings suggest that PS synthesis negatively regulates the magnitude and thresholds of BCR-induced Ca²⁺ elevation and downstream signaling in response to BCR ligation.

**Spontaneous BCR-derived Ca²⁺ signaling is a critical determinant of susceptibility to PTDSS1 inhibition**

In B cell lymphomas, BCR induces spontaneous Ca²⁺ elevation even in the absence of extracellular stimuli (Gururajan et al., 2006; Kume et al., 2019; Varano et al., 2017; Ziegler et al., 2019). In time-lapse imaging using YC3.6, a genetically encoded FRET-based Ca²⁺ indicator, we confirmed that autonomous elevation of cytosolic Ca²⁺ levels was observed only in B cell lymphoma expressing the components of the BCR including Ramos, but not in BCR-negative lymphoma or solid-tumor cell lines (Fig. 6 A). Intriguingly, the frequency (Fig. 6, B and C) and amplitude (Fig. 6 D) of spontaneous Ca²⁺ spikes and basal Ca²⁺ level (Fig. 6 E) were higher in a PTDSS1-KO Ramos clone than in parental Ramos cells. Treatment with PLCγ2 inhibitor suppressed the Ca²⁺ spikes in the PTDSS1-KO clone (Fig. 6 F), indicating that the Ca²⁺ spikes were due to spontaneous BCR activation. To determine if hyperactivation of continuous BCR signaling leads to cell death, we generated Ramos lines deficient in BCR signaling components (Fig. S2 H) and evaluated their susceptibility to PTDSS1i. Previous reports revealed that gene KO of BCR components does not affect the growth of Ramos cells in the non-competitive condition, although the cells are expected to lose fitness (He et al., 2018). In CD79B-KO Ramos cells, the efficacy of PTDSS1i was strikingly lower than that in parental Ramos cells (Fig. 6 G). KO of CD19,

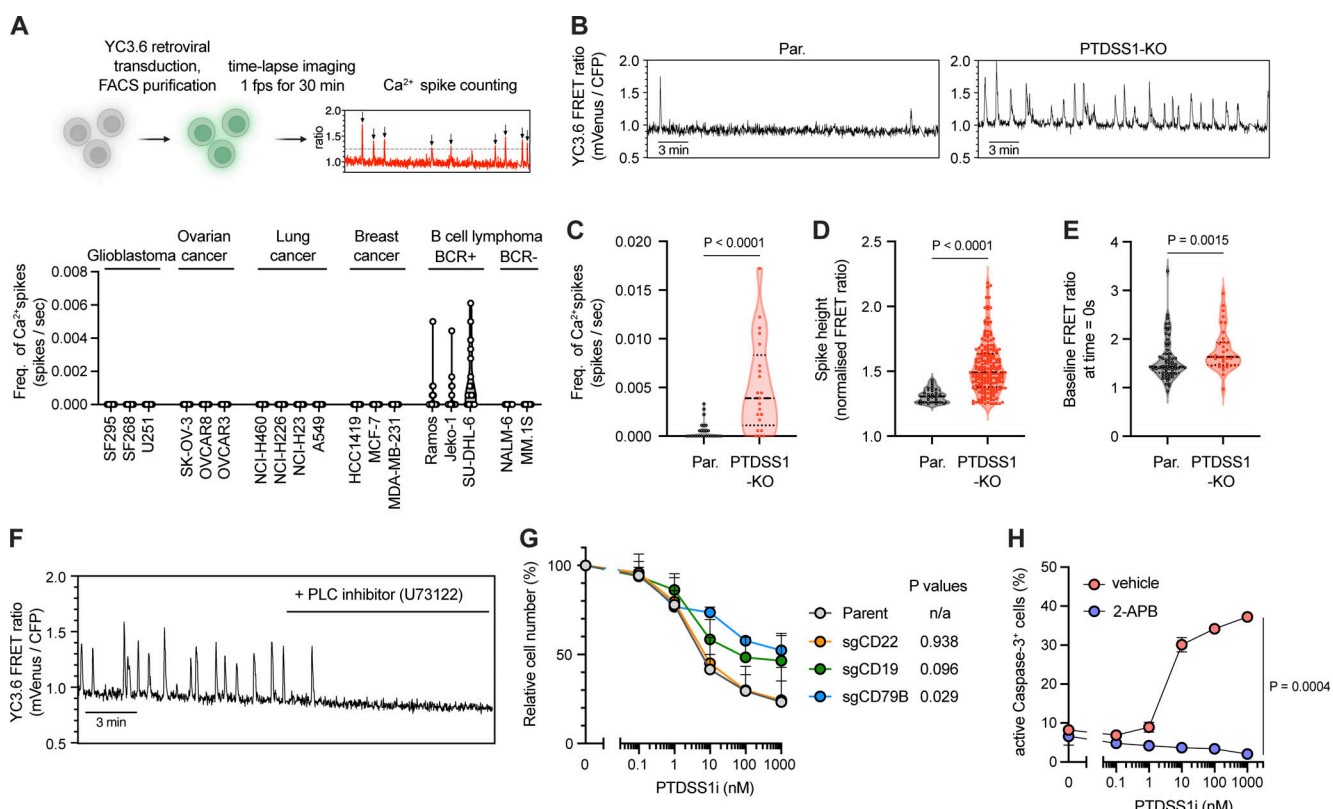

Figure 6. **BCR is a critical determinant of susceptibility to PTDSS1 inhibition. (A)** Spontaneous $Ca^{2+}$ elevation (mVenus/CFP increased to 1.25-fold over basal) in cancer cell lines transduced with YC3.6-expressing retroviral vector. 15–20 cells in the field were analyzed and plotted for each cell line. **(B–E)** Spontaneous $Ca^{2+}$ elevation in parental or PTDSS1-KO Ramos cells recorded as in B. Frequency of $Ca^{2+}$ spikes (C), spike height (D), and baseline FRET ratio (E) are shown. Each dot indicates individual cells in C and E or individual peaks in D. Statistical P values based on Mann–Whitney $U$ test are shown. **(F)** Suppression of spontaneous $Ca^{2+}$ elevation in PTDSS1-KO Ramos cells by pan-PLC inhibitor (U73122). **(G)** Susceptibility of KO Ramos cell lines to PTDSS1i. Data are presented as a percentage of the vehicle control value (mean + SD of three independent experiments). Statistical P values based on Welch's $t$ test are shown. The absolute cell concentrations (one million cells per ml) in the absence of PTDSS1i are 1.08 ± 0.07 for the parent, 0.99 ± 0.11 for sgCD22, 1.07 ± 0.11 for sgCD19, and 1.06 ± 0.23 for sgCD79B. **(H)** Reduced apoptosis in 2-APB-treated Ramos cells in the presence of PTDSS1i. Data are presented as a percentage of active caspase-3-positive cells (mean ± SD of three independent experiments). Statistical P values based on Welch's $t$ test are shown.

which localizes to the proximal region of the BCR and amplifies the BCR signal by enhancing PI3K-mediated $PI(3,4,5)P_3$ production (Otero et al., 2001; Tuveson et al., 1993), also attenuated the efficacy of PTDSS1i, whereas KO of CD22, a negative regulator of the BCR signal (Nitschke et al., 1997), did not change the efficacy. Moreover, we found that treatment with 2-aminoethoxydiphenyl borate (2-APB), which blocks both $Ca^{2+}$ release via $IP_3$ receptor and $Ca^{2+}$ entry via SOCE (Maruyama et al., 1997; Wei et al., 2016), significantly rescued the apoptotic cell death induced by PTDSS1i treatment (Fig. 6 H). These findings suggest that aberrant $Ca^{2+}$ elevation derived from continuous BCR signaling is induced by insufficient PS synthesis and is cytotoxic in B cell lymphoma.

## PTDSS1 controls the PI4P pool at the PM to regulate BCR-derived signaling

Inhibition of PTDSS1 did not enhance the surface expression of BCR signaling components, including coreceptors CD19 and CD22 (Fig. S2 I), the phosphorylation of downstream proteins in response to BCR ligation (Fig. S2 J), or the subcellular localization of phosphorylated PLCγ2 (Fig. S2 K); however, it resulted in elevated production of inositol monophosphate ($IP_1$), a

measurable metabolite of $IP_3$ (Fig. 7 A). Therefore, we examined the impact of PTDSS1 inhibition on phosphoinositide metabolism, which generates the PLCγ2 substrate $PI(4,5)P_2$ and its precursor PI4P. PTDSS1i treatment increased PIP, which mostly mirrors the PI4P level, in Ramos cells (Fig. 7 B). We confirmed the increased localization of the ectopically expressed PI4P probe (EGFP-2xP4M$^{SidM}$) at the PM in PTDSS1i-treated cells (Fig. 7 C). By contrast, PTDSS1 inhibition significantly weakened the PM localization of the PS probe (EGFP-2xPH$^{evectin2}$; Fig. S3 A). PTDSS1i treatment also increased $PIP_2$ at the PM (Fig. S3 B) and in the cell (Fig. S3 C), which mostly mirrors $PI(4,5)P_2$, but to a lesser extent than PIP (Fig. 7 B and Fig. S3 C). PTDSS1-KO Ramos clone similarly showed a significant increase in total PIP (Fig. S3 D). Interestingly, the total $PIP_2$ level was not significantly elevated in the PTDSS1-KO clone; nevertheless, the clone showed a significantly enhanced $IP_1$ production and $Ca^{2+}$ response (Fig. 5 B and Fig. S3 E). These observations suggest that PTDSS1 inhibition primarily increased the level of PI4P. Pharmacological inhibition of PI4KIIIα, which converts PI to PI4P in the PM, caused an acute reduction of PM localization of the PI4P probe in PTDSS1i-treated cells but did not affect that of the $PI(4,5)P_2$

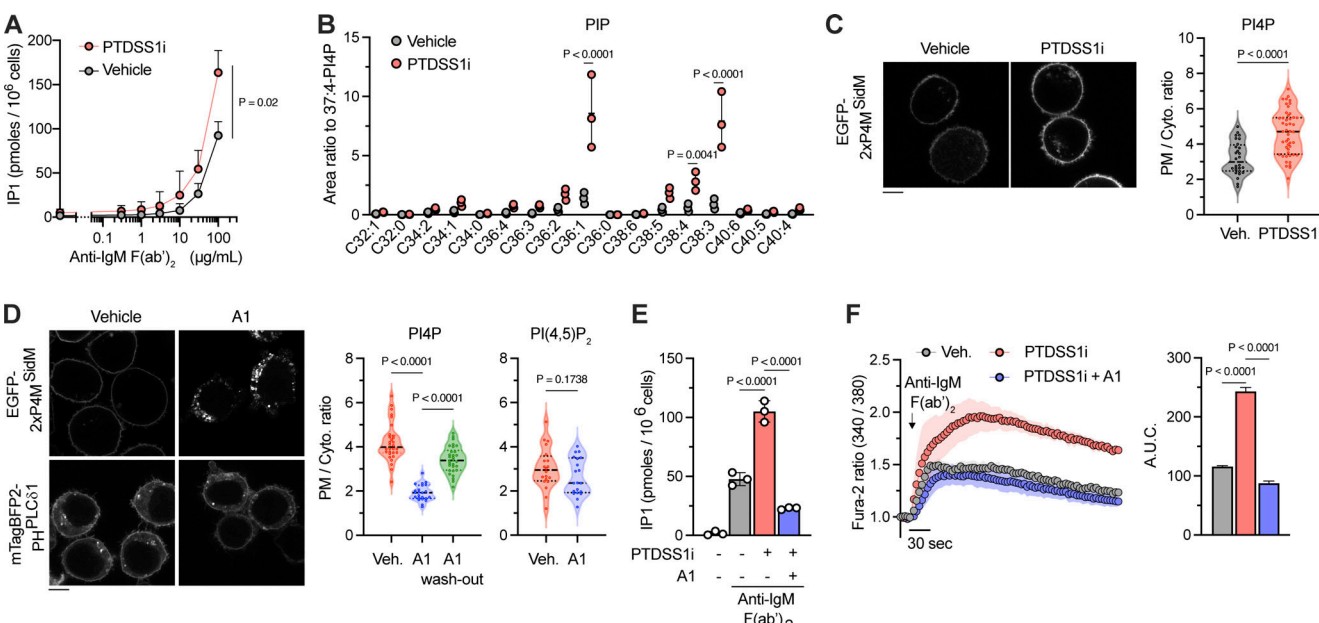

Figure 7. **PTDSS1 controls the PI4P pool at the PM to regulate BCR-derived signaling. (A)** IP$_1$ production in Ramos cells. The cells were cultured for 3 d with 100 nM PTDSS1i and then stimulated with the indicated concentrations of anti-IgM F(ab')$_2$ in the presence of LiCl. Data are presented as the mean + SD of three independent experiments. Statistical P values based on Welch's *t* test are shown. **(B)** PIP levels in Ramos cells cultured for 3 d with vehicle or 100 nM PTDSS1i. Data are presented as the mean ± SD of three independent experiments with a dot plot of values for the individual experiments. Statistical P values based on one-way ANOVA followed by Šídák's multiple comparisons test are shown. **(C)** Intracellular distribution of PI4P in vehicle- or PTDSS1i-treated Ramos cells. Ramos cells stably expressing EGFP-2xP4M$^{SidM}$ were cultured for 24 h with 100 nM PTDSS1i and then fluorescent images were analyzed using confocal microscopy (left). Fluorescence intensity ratio of PM to cytosol (PM/Cyto.) was quantified and dot-plotted for each cell (right). Statistical P values based on Mann–Whitney *U* test are shown. The scale bar represents 5 μm. **(D)** The effects of PI4KIIIα inhibitor (A1) on the intracellular distribution of PI4P and PI(4,5)P$_2$ in PTDSS1i-treated Ramos cells. Ramos cells stably expressing EGFP-2xP4M$^{SidM}$ or mTagBFP2-PH$^{PLCδ1}$ were cultured for 24 h with 100 nM PTDSS1i followed by treatment with 100 nM A1 for 60 min and then fluorescent images were analyzed using confocal microscopy (left). Fluorescence intensity ratio of PM to cytosol (PM/Cyto.) was quantified and dot-plotted for each cell (right). Statistical P values based on one-way ANOVA followed by Kruskal–Wallis multiple comparisons test are shown. Scale bar represents 5 μm. **(E and F)** The effects of A1 treatment on enhanced BCR signaling in PTDSS1i-treated Ramos cells. PTDSS1i-treated cells pre-treated with 100 nM A1 for 60 min were stimulated with 20 μg/ml anti-IgM F(ab')$_2$ and then IP$_1$ production (E) and Ca$^{2+}$ responses (F) were analyzed. Data are presented as the mean ± SD of three independent experiments. Statistical P values based on one-way ANOVA followed by Šídák's multiple comparisons test are shown.

probe (Fig. 7 D). PM localization of the PI4P probe was reversed at 30 min after the wash-out of the PI4KIIIα inhibitor (A1), suggesting the rapid turnover of the PI4P pool at the PM. Depletion of PM PI4P by treatment with A1 significantly attenuated the elevation of both BCR-derived IP$_1$ production and Ca$^{2+}$ response in PTDSS1i-treated cells (Fig. 7, E and F). Collectively, these findings suggested that PS synthesis by PTDSS1 primarily controls the PM PI4P pool, thereby promoting the downstream flow of PI(4,5)P$_2$ and PLCγ2-derived Ca$^{2+}$ response.

In contrast to PI4P, PM localization of PI(3,4,5)P$_3$ probe (PH$^{Akt1}$-EGFP) was slightly reduced in PTDSS1i-treated cells (Fig. S3 F). Consistently, we observed a slight decrease in cell surface expression of CD19 (Fig. S3 G), which tonically activates PI(3,4,5)P$_3$ production by PI3K and thereby downstream effectors such as Akt to promote the survivability of Burkitt's lymphoma. Nevertheless, we did not observe a significant decrease in phosphorylation of Akt in both basal and BCR-stimulated conditions in PTDSS1i-treated cells (Fig. S3 G).

### Lipid transporters mediate PI4P regulation by PTDSS1

Inhibition of PTDSS1 did not affect the mRNA expression of phosphoinositide kinases and phosphatases (Fig. S4 A), or the

subcellular localization of ectopically expressed PI4P-producing enzymes (PI4KIIα, PI4KIIβ, PI4KIIIα, and PI4KIIIβ) and PI4P-degrading enzyme, SACM1L (Fig. S4, B–F). Two molecules that might plausibly link PS synthesis to PIP metabolism are ORP5 and ORP8 (Sohn et al., 2018). Both phospholipid exchangers are ER transmembrane proteins but are also located in the contact site between the PM and the ER via the pleckstrin homology (PH) domain that binds PI4P or PI(4,5)P$_2$ in the PM. ORP5/8 transfers PI4P from the PM to the ER where it can be rapidly dephosphorylated to PI by SACM1L (Zewe et al., 2018), thus controlling PI4P level at the PM (Fig. 8 A). While both mCherry-fused ORP5 and ORP8 were mainly localized in the ER, they were drastically recruited to the PM in response to PTDSS1 inhibition (Fig. S4, G and H). We confirmed the colocalization of mCherry-ORP8 with GFP-MAPPER which probes the PM–ER contact sites (Chang et al., 2013; Fig. S4 I). The recruitment of mCherry-OPR8 was rapidly reversed by treatment with A1 (Fig. 8 B), suggesting that ORPs translocate to the PM–ER contact sites by sensing the elevation of PM PI4P in PTDSS1i-treated cells. To examine the role of ORP5 and ORP8 in the regulation of BCR-induced Ca$^{2+}$ response, we generated ORP5/8–double-knockout (DKO) Ramos cells (Fig. S5, A and B). We observed a

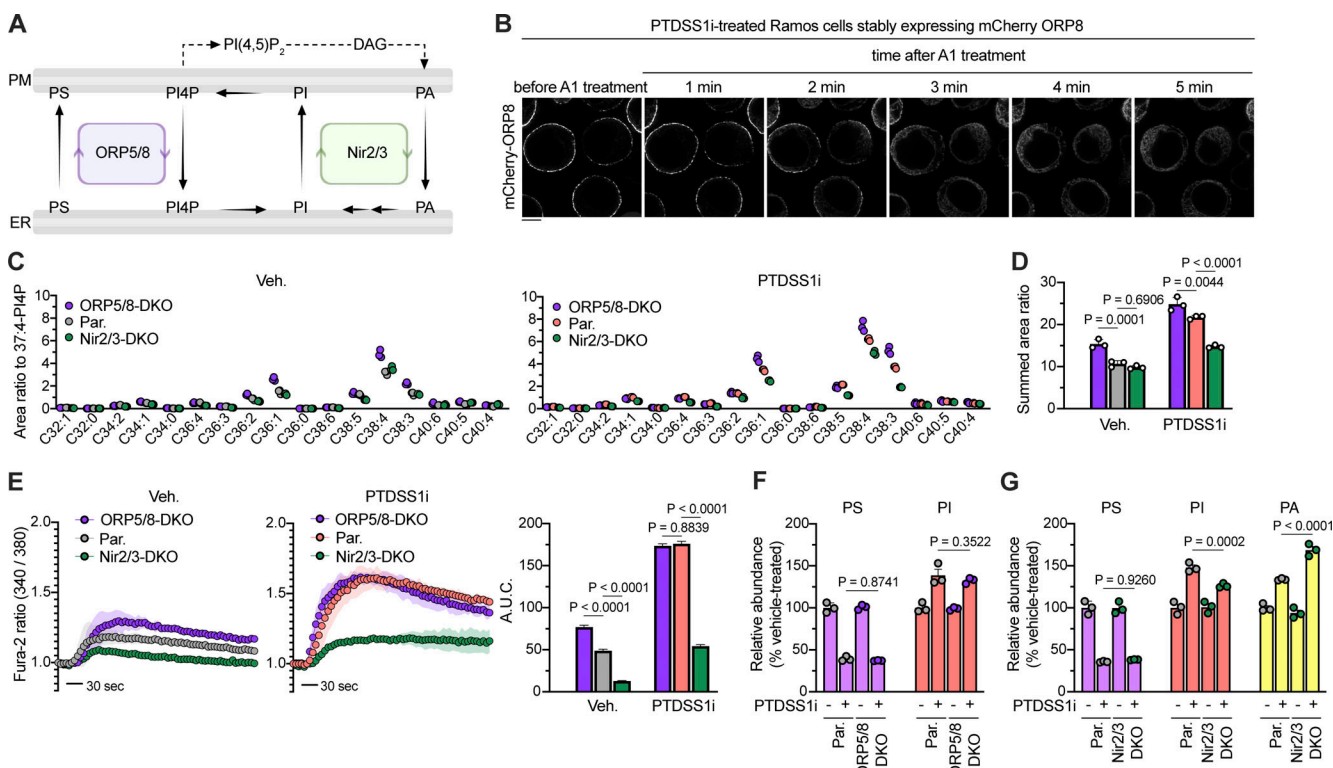

Figure 8. **Lipid transporters mediate BCR regulation by PTDSS1. (A)** Schematic of phospholipids transfer at the ER–PM contact site. **(B)** The effect of A1 on intracellular localization of mCherry-ORP8 in PTDSS1i-treated Ramos cells. Ramos cells stably expressing mCherry-ORP8 were cultured for 24 h with 100 nM PTDSS1i and then treated with 100 nM A1. Fluorescent images were analyzed at the indicated time points using confocal microscopy. Scale bar represents 5 μm. **(C and D)** PIP levels in ORP5/8-DKO or Nir2/3-DKO Ramos cells in the absence (left) or presence (right) of 100 nM PTDSS1i. Data are presented as a dot plot of values for the individual biological replicates. Statistical P values based on one-way ANOVA followed by Šídák's multiple comparisons test are shown. **(E)** BCR-induced Ca²⁺ responses of ORP5/8-DKO or Nir2/3-DKO Ramos cells in the absence (left) or presence (middle) of 100 nM PTDSS1i. Ramos cells pretreated with or without 100 nM PTDSS1i for 24 h were stimulated with 20 μg/ml anti-IgM F(ab')₂. The values for area under curve (A.U.C.) are shown (right panel). Data are presented as the mean ± SD of three independent experiments. Statistical P values based on one-way ANOVA followed by Šídák's multiple comparisons test are shown. **(F and G)** PL levels in ORP5/8-DKO or Nir2/3-DKO Ramos cells in the absence or presence of 100 nM PTDSS1i. Relative abundance for PS, PI, and PA are presented as a percentage of the values of the vehicle-treated control group. Each dot indicates the value for the individual biological replicates. Statistical P values based on one-way ANOVA followed by Šídák's multiple comparisons test are shown.

significant elevation of PIP in ORP5/8-DKO cells (Fig. 8, C and D). Consistently, ORP5/8-DKO cells showed enhanced Ca²⁺ response to BCR ligation (Fig. 8 E). Interestingly, however, the effects of ORP5/8-DKO were modest compared with those of PTDSS1 inhibition (Fig. 8, C–E). Moreover, PTDSS1 inhibition still increased PIP levels and Ca²⁺ response in ORP5/8-DKO cells, suggesting that the additional mechanism also contributes to PIP regulation by PTDSS1.

PTDSS1 inhibition also increased the levels of PI, which was not affected by ORP5/8-DKO (Fig. 8 F). Limiting the PI4P synthesis by PI4KIIIα inhibitor significantly attenuated BCR-derived responses in PTDSS1i-treated cells (Fig. 7, E and F), suggesting that the elevated PI potentially contributes to enhanced BCR signaling. Two members of membrane-associated PI transfer proteins, Nir2 (PITPNM1) and Nir3 (PITPNM2), mediate the countertransport of PA from the PM to the ER and PI from the ER to the PM (Chang and Liou, 2015; Kim et al., 2015). The Nir2/3 machinery couples to PI synthase and thus functions to maintain the PI level at the PM under both steady-state and PLC-activated conditions (Fig. 8 A). To examine whether the Nir2 and Nir3 contribute to the PI increase under PTDSS1i

inhibition, we generated Nir2/3-DKO Ramos cells (Fig. S5, C and D). While Nir2/3-DKO did not affect the PI levels under basal conditions, it impaired the elevation of PI under PTDSS1 inhibition and induced the concomitant accumulation of PA (Fig. 8 G). The elevation of PIP levels in response to PTDSS1 inhibition was also impaired in Nir2/3-DKO cells (Fig. 8, C and D). Consistently, we observed the attenuated Ca²⁺ response to BCR activation, even under the PTDSS1 inhibition (Fig. 8 E). These findings indicate that Nir2/3 machinery mediates the increase in PI and thus also PIP under PTDSS1 inhibition to promote BCR-derived Ca²⁺ response.

## Loss of PE contributes to PTDSS1i-induced cytotoxicity

PTDSS1i-treated Ramos cells also showed significant loss of PE, which potentially contributes to the severe cytotoxicity of PTDSS1 inhibition. Treatment with an exogenous ethanolamine (Etn) completely rescued the reduction of PE and PS and elevation of PI, and thus PTDSS1i-induced cytotoxicity in parental Ramos cells (Fig. 9, A and B). In PTDSS2-KO Ramos cells, PE, but neither PS nor PI, was selectively rescued by Etn treatment. In this condition, importantly, the cytotoxicity of PTDSS1i

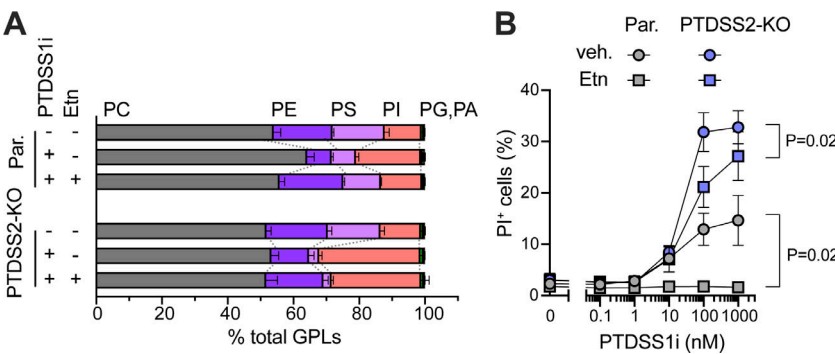

Figure 9. **Decreasing PE level contributes to the cytotoxicity of PTDSS1i-treated Ramos cells. (A and B)** Major phospholipid composition (A) and dead cell accumulation (B) of parental Ramos cells and PTDSS2-KO clone cultured in the indicated condition. The cells were cultured in the medium containing PTDSS1i (100 nM) and/or ethanolamine hydrochloride (0.1 mM) for 96 h, and then subjected to the analysis. Statistical P values examined based on Welch's *t* test are shown.

treatment was partially attenuated. Therefore, alterations not only of PIP but also of PE contribute to the cytotoxicity of PTDSS1 inhibition.

## Discussion

Using a recently developed PTDSS1 inhibitor, we screened 47 cancer cell lines for survivability under the restriction of PS synthesis and identified B cell lymphoma as a highly PS-dependent cancer type. In the absence of sufficient PS synthesis, an imbalance of the phospholipidome makes the BCR hyper-responsive, resulting in a dysregulated $Ca^{2+}$ response and massive cell death (Fig. 10).

BCR-positive lymphomas showed a transient, but substantial, elevation of cytosolic $Ca^{2+}$ (Fig. 6 A), which mirrors spontaneous BCR activation (Kume et al., 2019). This hallmark of B cell–derived lymphoma is considered an oncogenic and prosurvival signal and underlies the rationale for using BCR inhibitors to target downstream kinases such as BTK, SYK, and PI3Kδ (Burger and Wiestner, 2018). However, an extensive $Ca^{2+}$ signal due to BCR ligation triggers BCR activation-induced cell death in B cell lymphoma as well as normal B cells, whereby the immune system destroys hyper- or autoreactive B cells through a series of developmental stages (Sandel and Monroe, 1999). Therefore, the BCR-derived $Ca^{2+}$ signal is a double-edged sword for B cell lymphoma survival, and the BCR is equipped with multiple regulatory mechanisms to tune its amplitude, sustainability, and frequency of activation.

PIPs are extensively characterized phospholipids that are closely related to the molecular regulation of BCR signaling. $PI(4,5)P_2$ is not only a substrate for $IP_3$ generation but also a regulator for the formation of BCR microclusters by organizing the F-actin cytoskeleton, which constrains the motility of BCR on the PM (Wang et al., 2017). Therefore, the sizes of the phosphoinositides pool at the PM significantly impact BCR activity and are dynamically maintained by multiple regulators, including phosphoinositide kinases (PI4Ks, PIP5Ks, and PI3K) and phosphatases (INPP5B and PTEN; Droubi et al., 2022; Luo et al., 2019; Saci and Carpenter, 2005; Saito et al., 2003). Our data demonstrate that PS synthesis substantially controls the PI4P levels via membrane contact-based lipid transfer machinery, thereby the flow of downstream BCR signaling. It is reasonable that ORP5/8-DKO cells partially phenocopies PTDSS1i-treated cells because efficient PI4P transfer by ORPs largely depends on the presence of PS in the acceptor membrane (Ikhlef et al., 2021). Thus, the PI4P-PS countertransport mediated by PTDSS1

and ORP5/8 functions as a negative regulator for the PM PI4P pool and BCR signaling. Moreover, PTDSS1 inhibition leads to the concomitant increase in PI level in a Nir2/3-dependent manner. This finding suggests the unexpected link between PS and the so-called "PI-cycle," where PLC/DAG kinase-derived PA is efficiently recycled to form CDP-DAG, PI, and thus also PIPs. Therefore, both a deceleration of PI4P-PS countertransport by ORP5/8 and an acceleration of the PI-cycle by Nir2/3 contribute to the increased PIP and BCR hyperactivation under PTDSS1 inhibition (Fig. 10). While PTDSS1 inhibition did not affect the morphology of the perinuclear ER as judged by anti-KDEL antibody staining (Fig. S2 A), it increased the recruitment of ORP8 and MAPPER to PM–ER contact sites through PI4P and $PI(4,5)P_2$ accumulation in the PM (Fig. 8 B and Fig. S4 I). Unlike ORP5/8, the subcellular localization of Nir2/3-EGFP was not overtly affected by PTDSS1 inhibition (Fig. S5, E and F). Although the elevated PA observed in some cancer cell lines also potentially enhances the association of Nir2/3 with the PM, it is still unclear how PTDSS1 inhibition activates the Nir2/3-mediated PI-cycle. Furthermore, PTDSS1 inhibition still increases the PI level even in Nir2/3-DKO cells (Fig. 8 G), suggesting an additional Nir2/3-independent link between PS and PI metabolism. Interestingly, the increase in PI levels under PTDSS1i treatment was a feature observed in BCR-positive, or analogously TCR-positive, lymphoma lines but not in BCR-negative lymphoma lines or most solid-cancer lines. This suggests that the close link between PS synthesis and phosphoinositide metabolism forces BCR-positive lymphoma to be highly dependent on PS synthesis to tune the BCR signal. Considering that RTX- and PTDSS1i-induced apoptosis share a similar mechanism, our findings suggest that PS synthesis may be a critical vulnerability of malignant B cell lymphomas that can be targeted pharmacologically.

The exchange of PS/PIPs between organelles has diverse biological functions. PI4KIIIα is required for PS transfer from the ER to the PM via ORP5 and ORP8 and ensures PM targeting of oncogenic K-Ras, whose polybasic domain strongly interacts with PS (Kattan et al., 2021). Therefore, PI4KIIIα inhibition efficiently blocks the proliferation of cancers harboring oncogenic K-Ras mutation. In addition, PI4KIIα rapidly accumulates on damaged lysosomes, where it produces PI4P that is exchanged for PS from the ER via ORP9, ORP10, ORP11, and OSBP to repair the lysosomal membrane (Tan and Finkel, 2022). Furthermore, the exchange of PS/PIPs also occurs between ER and lipid droplets via ORP5 (Du et al., 2019). All these reported functions share the feature that PIP synthesis is a driver of

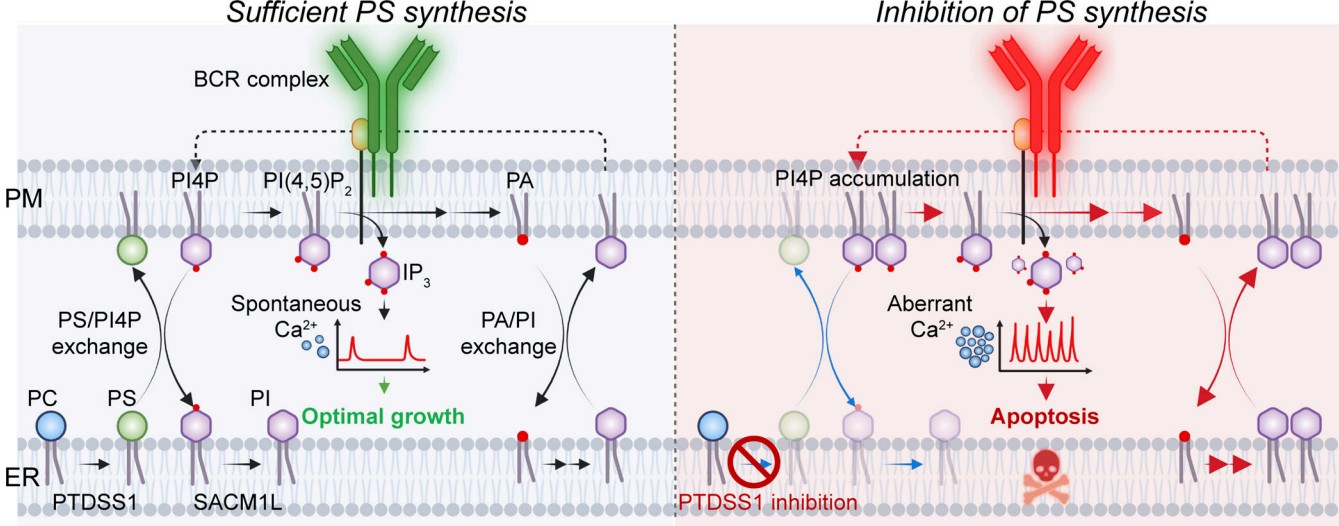

Figure 10.  **Mechanism of the regulation of BCR signaling by PTDSS1.** In B cell lymphoma, the BCR complex continuously activates downstream PLCγ2, which produces IP3 by cleaving PI(4,5)P2 in the PM, causing spontaneous cytosolic Ca2+ elevation. PA derived from the PLC reaction stimulates PA/PI exchange via Nir2/3 and is recycled to form PI and PI4P to maintain the PM phosphoinotides pool positively (PI-cycle). By contrast, PS synthesis by PTDSS1 in the ER is coupled to PS/PIP exchange via ORP5/8 and controls the PM PI4P pool negatively. Under optimal positive and negative regulations, the spontaneous Ca2+ supports the survival and proliferation of B cell lymphoma (left). PTDSS1 inhibition causes both deceleration of PS/PI4P exchange (blue arrow) and acceleration of PI-cycle (red arrow), leading to aberrant BCR-derived Ca2+ signaling and, ultimately, cell death (right).

countertransport to maintain PS levels in target organelles. Experiments with a gain-of-function PTDSS1 mutant (Sohn et al., 2016) revealed that PS synthesis is another factor that governs phosphoinositide levels, although the biological significance of this mode of phosphoinositide regulation has not been determined. Our findings suggest that the control of phosphoinositide levels by PS synthesis is important in B cell lymphoma biology.

PTDSS1 inhibition causes a decrease in PE levels in a wide range of cancer cells. This reduction is possibly due to reduced decarboxylation of PS and can be compensable by a CDP–ethanolamine pathway (Fig. 9 A). The cytotoxicity of PTDSS1i treatment was partially rescued by co-treatment with an exogenous ethanolamine in PTDSS2-KO Ramos cells, which have a normalized PE level but aberrant PS and PI levels. Reduction of the PE/PC ratio causes loss of Ca2+ uptake into the ER (Fu et al., 2011), and this mechanism links the altered phospholipid metabolism in obesity to Ca2+-induced cell stress. Therefore, alterations not only of PIPs but also of PE potentially contribute to the cytotoxicity of PTDSS1 inhibition in B cell lymphoma. Thus, PS synthesis balances the phospholipid environment that fine-tunes BCR signaling, although the direct contribution of PS to BCR regulation remains to be clarified.

Loss of PTDSS1 does not reduce the PS levels in several tissues of PTDSS1-deficient mice, suggesting that PTDSS2 can compensate for the PS synthesis from PE in vivo (Arikketh et al., 2008). However, this is not the case for B cell lymphomas, which greatly required PS synthesis by PTDSS1, but not PTDSS2, for their growth in both in vitro culture and in vivo. One possible explanation for the high PTDSS1 dependency is that B cell lymphomas may distinguish the difference in the acyl-chain species of PS synthesized by PTDSS1 and PTDSS2. A previous

report reveals that PTDSS2 efficiently synthesizes docosahexaenoic acid (DHA)-containing PS from DHA-containing PE (Kimura and Kim, 2013), which is substantially present in the various tissues in vivo. Consistent with this, whereas the levels of C34:1- or C36:1-PS were reduced, the levels of DHA-containing PS such as C40:6-PS significantly increased in PTDSS1-KO Ramos cells. Therefore, the inactivation of PTDSS1 causes imbalances not only in the composition of phospholipid head classes but also in the acyl-chain composition of PS. A recent report revealed that ORP8 transfers the acyl-chain species of PS differently (Ikhlef et al., 2021). Therefore, the imbalance in the acyl-chain composition of PS in the condition of PTDSS1 inactivation may affect the efficiency of the exchange of PS/PIPs in B cell lymphomas.

Two recent studies demonstrated that PTDSS1 is an attractive target for combating specific cancers. One study revealed that breast cancer cells expose high levels of ether-linked PS species on their surface that activate the PS receptor MerTK on tumor-associated macrophages (TAMs; Sekar et al., 2022). This tumor-to-TAM interaction augments the proliferation and immune-suppressive functions of TAMs, shaping the tumor microenvironment in vivo. Consistent with that, PTDSS1 expression is negatively correlated with the survival of patients with breast carcinoma (Sekar et al., 2022). Another study suggested that cancer cell populations with PTDSS2 deletion are highly susceptible to PTDSS1 inhibition (Yoshihama et al., 2022). This strategy is based on findings that PTDSS1/2 double KO, but neither single KO, is lethal in mice (Arikketh et al., 2008). Moreover, PTDSS1i exposure induces immune potentiation by stimulating HMGB1 secretion to activate the antitumor responses of dendritic cells (Yoshihama et al., 2022). Our screen revealed that B cell lymphoma is highly dependent on PTDSS1 despite intact PTDSS2

expression. Impacts on phospholipid networks underly the specific mechanism of action against B cell lymphomas and potentiate the clinical drug rituximab, suggesting a novel strategy for antilymphoma therapy development.

## Materials and methods

### Antibodies and reagents
Antibodies and reagents were purchased by vendors and used at the indicated dilution or concentration as follows: rabbit monoclonal anti-PTDSS1 (1:1,000, Cat# ab157222; Abcam), mouse monoclonal anti-α-tubulin (1:2,000, Cat#T6199; Sigma-Aldrich), rabbit polyclonal anti-Nir2 antibody (1:500, Cat#HPA003978; Sigma-Aldrich), rabbit polyclonal anti-ORP8 antibody (1:500, Cat#99069; Abcam), mouse monoclonal anti-KDEL antibody (1: 1,000, Cat#ADI-SPA-827; Enzo Life Sciences), mouse monoclonal anti-GM130 antibody (1:500, Cat#610822; BD Biosciences), anti-GS28 antibody (1:500, Cat#611184; BD Biosciences), rabbit polyclonal anti-COXIV antibody (1:500, Cat#PM063; MBL), mouse monoclonal anti-SNX1 antibody (1:500, Cat#611482; BD Biosciences), mouse monoclonal anti-TfnR antibody (1:500, Cat#ab38171; Abcam), FITC-conjugated mouse monoclonal anti-human IgM (1:200, Cat# 314506; BioLegend), PE-conjugated mouse monoclonal anti-human CD79B (1:500, Cat# 341404; BioLegend), PE-conjugated mouse monoclonal anti-human CD19 (1:1,000, Cat# 555413; BD Biosciences), PE-conjugated mouse monoclonal anti-human CD79B (1:500, Cat# 341404; BioLegend), APC-conjugated mouse monoclonal anti-human Ig light chain λ (1:200, Cat# 316610; BioLegend), Alexa Fluor 647-conjugated mouse monoclonal anti-human CD22 (1:500, Cat# 302517; BioLegend), PE-conjugated mouse monoclonal anti-pY759-PLCγ2 (1: 6, Cat# 558490; BD Biosciences), PE-conjugated mouse monoclonal anti-pY84-BLNK (1:6, Cat# 558442; BD Biosciences), PE-conjugated mouse monoclonal anti-pY352-Syk (1:20, Cat# 683704; BioLegend), goat anti-human IgM F(ab')$_2$ (Cat# 397302; BioLegend), HRP-conjugated donkey anti-rabbit IgG F(ab')$_2$ (1:10,000, Cat#NA9340; Cytiva), HRP-conjugated sheep anti-mouse IgG F(ab')$_2$ (1:10,000, Cat#NA9310; Cytiva), NucView488 (1 μmM, Cat#10402; Biotium), U73122 (10 μmM, Cat#ab120998; Abcam), and GSK-A1 (100 nM, Cat#SML2453; Sigma-Aldrich). The specific antibody against human PTDSS2, PTDSS1 inhibitors (DS68591889 and DS55980254), and Ibrutinib were prepared in Daiichi Sankyo Co., Ltd. We used DS68591889 as a PTDSS1i for most experiments unless otherwise specified. The tFucci(CA)2/pCSII-CMV (Cat# RDB15452) and YC3.6/pcDNA3 (Cat# RDB15135) were provided by the RIKEN BRC through the National BioResource Project of the MEXT, Japan (Nagai et al., 2004; Sakaue-Sawano et al., 2017). pSpCas9(BB)-2A-GFP (PX458) was a gift from Feng Zhang (Ran et al., 2013). The lentiviral vectors were developed by the late Dr. Hiroyuki Miyoshi. psPAX2 and pMD2.G were gifts from Didier Trono.

### Cell culture
The culture media, culture conditions, and RRIDs for 47 cell lines are described in Table S1. Representatively, Ramos cells were obtained from JCRBCB (Japanese Collection of Research Bioresources Cell Bank) and were maintained at 37°C in complete RPMI1640 supplemented with 10% heat-inactivated fetal bovine serum (FBS), 100 U/ml penicillin and 100 μg/ml streptomycin, and 292 μg/ml L-glutamine. For comparison of growth, survival, and phospholipid composition among PTDSS1-KO or PTDSS2-KO lines, Ramos, HeLa, SU-DHL-6, and A549 cells were cultured in complete RPMI1640. In some experiments (Fig. 5, B and C), the cells were cultured in complete RPMI1640 supplemented with 20% FBS to reduce spontaneous apoptosis. Jeko-1-Luc cells were prepared by transfecting the firefly luciferase gene in Daiichi Sankyo RD Novare Co., LTD. and cultured in RPMI1640 supplemented with 10% heat-inactivated FBS.

### Cell-free PTDSS assay
The inhibitory potency of DS68591889 against cell-free PTDSS1 or PTDSS2 activity was examined following the same protocol as described previously (Yoshihama et al., 2022). Briefly, the membrane fraction of Sf9 expressing human PTDSS1 or PTDSS2 was mixed with the indicated concentrations of DS68591889 in reaction solutions containing 50 mM HEPES-NaOH pH 7.5, 5 mM CaCl$_2$, and 1 μCi/ml L-[$^{14}$C(U)-serine], and incubated at 37°C for 20 min. After stopping the reaction by adding 10 mM EDTA, the membrane fractions were trapped onto Unifilter-96 GF/C (PerkinElmer), washed, and then scintillation counts were measured using TopCount-NXT-HTS (PerkinElmer).

### Screening of human cancer cell lines using PTDSS1i
The cells cultured on a 24-well plate were incubated with the indicated concentrations of PTDSS1i. For phospholipid analysis, the cells were collected on day 2, washed once with PBS, and then suspended in methanol containing internal standards for PS, PC, PE, PI, PG, and PA. Each phospholipid level in methanol extracts was analyzed LC-MS/MS as described below. To standardize the variation of cell numbers among samples during collection, the ratio to PC was calculated for each phospholipid class (PS, PE, PI, PG, and PA) and used for data presentation. For cell growth assay, cells were collected on day 4 or 5 after PTDSS1i treatment, 70-μm-filtered, and then total cell numbers were determined by EC800 cell analyzer (SONY).

### Phospholipids measurement
Cells, typically cultured in a 24-well plate, were washed once with PBS and suspended in methanol containing internal standards for 100 nM dilauroyl-PS (di12:0-PS), 1 μM di12:0-PC, 1 μM di12:0-PE, 100 nM di12:0-PG, 100 nM di12:0-PA, 1 μM d18:1/C17:0-ceramide, 1 μM d18:1/C17:0-sphingomyelin, and 1 μM didecanoyl-diacylglycerol. Those purified lipids were purchased from Avanti Polar Lipids. The methanol-soluble fractions were obtained by centrifuge and analyzed by LC-MS/MS system. For PIPs' measurement, chloroform-soluble fractions were obtained, mixed with 17:0/20:4-PI, 17:0/20:4-PI4P, 17:0/20:4-PI(4,5)P$_2$, and 17:0/20:4-PI(3,4,5)P$_3$ as internal standards, and then subjected to phosphate methylation reaction using TMS-diazomethane as described previously (Shimanaka et al., 2022). LC-MS/MS system was composed of ultraperformance liquid chromatography (ACQUITY UPLC; Waters) and an ESI tandem quadrupole mass spectrometer (Xevo TQ-XS; Waters). The injected samples were separated on a reverse-phase column (L-column 3 C18, particle

size 2 µm, inner diameter 2.0 mm, length 100 mm; CERI) with the flow rate set to 0.3 ml/min at 45°C in a binary gradient system using acetonitrile/water (3:2, vol/vol) containing 10 mM ammonium formate as a mobile phase A and isopropanol/acetonitrile (9:1, vol/vol) containing 10 mM ammonium formate as a mobile phase B. The gradient condition was set as follows: initial condition (0 min), 5% B; 0–2 min, 5% B; 2–26.4 min linear gradient to 100% B; 26.4–36 min, 100% B; 36–36.1 min, linear gradient to 5% B; 36.1–40 min, 5% B. The capillary voltage, sampling cone voltage, and desolvation temperature were set to 2.5 kV, 30 V, and 400°C, respectively. The acyl-chain species of PC, PE, PS, PI, PG, PA, DAG, sphingomyelin, ceramide, and PIPs were detected using the multiple reaction monitoring (MRM) method, as reported previously. Raw data files were processed and analyzed using MassLynx software (Waters).

### Establishment of gene-KO cells, stably gene-expressing cells, and stably gene-knockdown cells

Ramos-derived KO cells of *PTDSS1*, *PTDSS2*, *CD22*, *CD19*, *CD79B*, *OSBPL5*, *OSBPL8*, *PITPNM1*, and *PITPNM2* were generated by CRISPR-Cas9 genome editing using the pSpCas9(BB)-2A-GFP (PX458) vector. DNA fragments targeting the genes (Table S2) were introduced into PX458 according to the original protocols. According to the manufacturer's procedure, Ramos cells were transfected with the construct using Neon Transfection System (Invitrogen). On day 2 after transfection, GFP-positive cells were isolated using SH800 cell sorter. For the generation of PTDSS1- or PTDSS2-KO clones, the isolated cells were subsequently seeded into a 96-well plate at a single cell per each well and expanded in RPMI1640 supplemented with 20% FBS. The knockout of PTDSS1, PTDSS2, OSBPL5, OSBPL8, PITPNM1, or PITPNM2 was confirmed by genomic direct sequencing or Western blot. For the generation of CD22, CD19, and CD79B-KO Ramos cells, sorted GFP-positive cells were expanded in bulk culture, stained with fluorescence-conjugated specific antibodies to each molecule, and then surface expression-negative bulk populations were enriched and used for the experiments. The same procedures were adopted for gene KO in SU-DHL-6, A549, or HeLa cells.

Ramos cells stably expressing YC3.6 were generated by retroviral transduction system using pMRX vector. The gene fragment amplified by PCR using specific primers (Table S2) was introduced into the pMRX digested with BamHI and EcoRI using NEBuilder (New England Biolabs) according to the manufacturer's protocol. The recombinant retroviruses were produced by cotransfection of the pMRX construct and pVSVG into the Plat-E cells using Lipofectamine 2000 transfection reagent (Invitrogen). Ramos cells were infected with the recombinant virus by plate centrifuge at 1,500 × *g* for 90 min, washed once, and expanded. After the expansion, stably transduced cells were sorted based on the fluorescence of genes using SH800 cell sorter.

Ramos cells stably expressing phospholipid probes, PI kinases, phosphatase, and transfer proteins fused with a fluorescent protein were generated by third-generation lentivirus system. Briefly, DNA fragments amplified by PCR using specific primers (Table S2) from the cDNA of Ramos cells were introduced into NheI/XbaI-digested tFucci(CA)2/pCSII-CMV using the TEDA cloning method. The resulting constructs were cotransfected

with psPAX2 and pMD2.G into 293FT cells, and the lentiviral supernatant was obtained 48 h after transfection. Ramos cells were infected with the supernatant as described above, and stably transduced cells were sorted based on the fluorescence of genes using SH800 cell sorter.

### Flowcytometry analysis

The suspended cells were stained for 30 min under the following conditions: propidium iodide (1 µg/ml), NucView488 (1 µM), FITC-conjugated anti-IgM antibody (1:200), PE-conjugated anti-CD79B antibody (1:200), PE-conjugated anti-CD19 antibody (1:500), or PE-conjugated anti-CD22 antibody (1:500). The stained cells were then analyzed by BDFACSLyric. For intracellular staining, Ramos cells were stimulated with the indicated concentrations of anti-IgM F(ab')$_2$ at 37°C for 4 min and then immediately fixed with a PFA-based fixation buffer (BioLegend) for 20 min. After membrane permeabilization using saponin-based wash buffer (BioLegend), fixed cells were stained for 1 h under the following conditions: PE-conjugated anti-phospho-PLCγ2 (1:12), PE-conjugated anti-phospho-ZAP70/Syk (1:40), PE-conjugated anti-phospho-BLNK (1:12), and PE-conjugated anti-phospho-Akt (1:12). After two washes, the stained cells were then analyzed by BDFACSLyric.

### Western blot

The cells were lysed in RIPA buffer supplemented with cOmplete protease inhibitors cocktail (Roche). After brief sonication and centrifuge, total protein concentrations were quantified based on the BCA assay (Thermo Fisher Scientific), followed by mixing with Laemmli's SDS sample buffer and boiling at 95°C for 5 min. For samples to detect PTDSS1 or PTDSS2, the boiling step was not performed to avoid the aggregation of those targets. Total cell lysates (typically 10 µg) were separated by Laemmli's SDS-PAGE system and transferred to a polyvinylidene difluoride membrane using a Criterion transblotter (Bio-Rad). After blocking with 10% skim milk, the membrane was immunoblotted with the indicated primary antibodies followed by HRP-conjugated anti-mouse or rabbit IgG F(ab')$_2$ and then visualized by immunostar LD chemiluminescence reagent (Wako) using FUSION SOLO.7s.EDGE imager (Vilber Bio Imaging).

### Quantitative RT-PCR

Total RNA was collected from Ramos cells cultured in the presence or absence of 100 nM PTDSS1i for 3 d using RNeasy Plus Mini kit (Qiagen) and transcribed into cDNA using High-Capacity cDNA Reverse Transcription Kit (Thermo Fisher Scientific). Realtime qRT-PCR was performed using the obtained cDNA as a template and gene-specific primers (Table S2). Data are analyzed by relative quantification based on the ddCT methods using *36B4* as a reference gene.

### Microscopy analysis

Ramos cells were cultured with vehicle or PTDSS1i for the indicated time. For immunocytochemical analysis, the cells were fixed with 3.75% paraformaldehyde, permeabilized with 0.1% TritonX-100, and then stained with specific primary antibodies against organelle markers followed by staining with Alexa Fluor

488-labeled secondary antibody. The fixed cells were suspended in PBS as an imaging medium and seeded onto a PLL-coated 4-well glass-bottom dish and observed using LSM980 laser scanning confocal microscopy system (Zeiss) consisting of Axio Observer 7 inverted microscope equipped with Plan-Apochromat 63×/1.4 objective lenses, a motorized stage, DefiniteFocus3 autofocusing device, and AiryScan2 super-resolution imaging detectors. All imaging procedures were performed under room temperature (typically 25–28°C) and operated with ZEN Blue software (Zeiss).

## Ca²⁺ measurement

For measuring the bulk $Ca^{2+}$ response, the cells were stained with Fura-2 (5 μM) in $Ca^{2+}$-free HBSS at 37°C for 45 min, followed by two washes with $Ca^{2+}$-free HBSS. The stained cells were seeded into a half-area 96-well plate with a black wall and clear bottom and then stimulated with the indicated compounds. Emission at 510 nm by either 340 or 380 nm excitation was recorded using FlexStation3 with Flex mode to calculate the fluorescence ratio. Alternatively, for measuring the $Ca^{2+}$ elevation at a single-cell resolution, cells were transduced with YC3.6-expressing retrovirus vectors as described above. The YC3.6-expressing cells suspended in HBSS containing 1% BSA were seeded onto a PLL-coated 4-well glass-bottom dish and observed using LSM980. The CFP fluorescence and mVenus fluorescence excited at 445 nm diode laser were simultaneously recorded every second for 30 min, and the ratio of mVenus (FRET) to CFP was calculated to correct baseline drift during imaging. All imaging procedures were performed under room temperature (typically 25–28°C) and operated with ZEN Blue software (Zeiss). In some experiments, the indicated compounds were added during imaging. Typically, >15 cells in the field were selected at random and subjected to analysis and data presentation.

## IP₁ measurement

Cellular $IP_1$ was measured using IP-One Gq kit HTRF (Cisbio) according to the manufacturer's protocol. Cells suspended in lithium chloride-containing stimulation buffer to inhibit the $IP_1$ degradation were seeded onto a 384-well OptiPlate (Perkin-Elmer) and then stimulated with the indicated concentrations of compounds at 37°C for 30 min. The cells were then lysed within the well by adding the HTRF-based detection reagents. The emission signals at 615 and 665 nm by excitation at 320 nm were measured using EnVision plate reader equipped with APC and europium filters (PerkinElmer). The absolute quantification was performed by interpolation using the predetermined $IP_1$ standard curve.

## Mice xenograft for B cell lymphomas

Male 5-wk-old, specific pathogen-free (SPF) BALB/cAJcl-nu (nu/nu) mice were obtained from CLEA Japan and maintained for at least a week prior to tumor injection. All mice were housed in climate-controlled (23°C) facilities with a 12-h light/12-h dark cycle. Parent, PTDSS1-KO, or PTDSS2-KO Ramos cells (20 million cells per 200 μl PBS) were subcutaneously injected into the left flank of mice. The tumor development was monitored from day 7 after injection by measuring the width (W) and length (L) of tumors using a vernier caliper, and tumor volumes (V) were calculated using the formula $V = (W^2 \times L)/2$. Mice were euthanized

at day 19, a humane endpoint predefined as the time at which either tumor width or length in each of the mice reaches 20 mm. This animal experiment was approved by the animal ethics committee of the University of Tokyo prior to its commencement and performed in accordance with approved protocols.

Female 6-wk-old, NSG mice were obtained from Charles River Laboratories Japan, Inc. The mice were housed in sterilized cages and maintained under specific pathogen-free conditions set at 23°C ± 2°C and 55% ± 10% humidity. The mice were intravenously injected with Jeko-1-Luc cells (one million cells per 200 μl D-PBS) and randomly grouped on the next day. From the day of grouping, the mice were orally administered with the indicated doses of PTDSS1i (DS55980254) or Ibrutinib once daily for 21 d. 0.5% methylcellulose was used as a vehicle solution. On day 21 after tumor injection, the mice were intraperitoneally injected with 150 mg/kg VivoGlo Luciferin (Promega), and the luminescence activity was evaluated using in vivo imaging system (IVIS 200 Image system; PerkinElmer Inc.) after 20 min of the injection. Total flux (p/s) and average radiance (p/s/cm²/sr) of luminescence activity were calculated with analysis software (Living Image Software ver.4.3.1; PerkinElmer Inc.). Mice that showed abnormal behavior (abnormal posture, abnormal walking, and paralysis) or mice with body weight loss of >25% were euthanized and regarded as dead. The survival rate was analyzed by Kaplan–Meier survival analysis. This experiment was approved by the Institutional Animal Care and Use Committee of Daiichi Sankyo Co., Ltd.

## Statistical analysis

Significant differences between the two groups were analyzed using Welch's $t$ test or Mann–Whitney $U$ test. Multiple comparisons of differences among every selected group were analyzed using one-way ANOVA followed by Šídák's multiple comparisons test. Dunnett's test was used for multiple comparisons of one control group with the other experimental groups. Significant differences in survival rates were analyzed using the log-rank test. All statistical analysis was performed using GraphPad Prism 9. No statistical methods were used to determine the sample size. Data distribution was assumed to be normal, but this was not formally tested. For the animal experiments, 6–10 mice were used, which should be sufficient to identify the effects among groups. No blinding method was applied to this study.

## Online supplemental materials

Fig. S1 shows the effects of PTDSS1-KO or PTDSS2-KO on the phospholipid levels in Ramos, SU-DHL-6, HeLa, and A549 cell lines. Fig. S2 includes the effects of PTDSS1 inhibition on the morphology of organelles, cell surface expression of BCR components, or BCR-induced phosphorylation of downstream molecules. Fig. S3 shows the effects of PTDSS1 inhibition on the intracellular distribution of PS and PIPs. Fig. S4 shows the effects of PTDSS1 inhibition on the expression or intracellular localization of PI-4-kinases, phosphatases, and transfer proteins. Fig. S5 describes the CRISPR-Cas9–mediated mutation in ORP5/8-DKO and Nir2/3-DKO clones. Table S1 lists the culture conditions and RRID accession for cancer cell lines used in this study. Table S2 lists the sequence for the oligonucleotides used in this study.

## Data availability

The data are available from the corresponding author upon reasonable request.

## Acknowledgments

We gratefully acknowledge Dr. Makoto Murakami and Dr. Mikihiko Naito for instrument and equipment support. The Ca²⁺ live-cell imaging experiment was performed with the help of the University of Tokyo and Carl Zeiss Collaboration Center (TZCC). This research was conducted in collaboration with the University of Tokyo and Daiichi Sankyo Co., Ltd.

This work was supported by grants from the Japan Society for the Promotion of Science KAKENHI, grant numbers 22K15272 and 21J00906 (to J. Omi); the Japan Agency for Medical Research and Development, grant numbers JP21gm1210013 (to N. Kono), and JP21gm0010004h9905 and JP22ck0106533h0003 (to J. Aoki); Japan Science and Technology Agency Moonshot R&D Program, grant numbers JPMJMS2023-11 (to J. Aoki) and JPMJMS2023-15 (to J. Aoki). T. Kato and Y. Yoshihama are employees of Daiichi Sankyo Co., Ltd.

Author contributions: J. Omi, N. Kono, and J. Aoki designed the research. J. Omi performed most of the experiments, analyzed and interpreted the data, and wrote the manuscript. T. Kato and Y. Yoshihama provided a key material (DS68591889 and DS55980254) and performed cell-free PTDSS assay and Jeko-1 xenograft experiments using PTDSS1i. K. Sawada established and analyzed HeLa cell lines deficient for PS synthase. J. Aoki and N. Kono provided critical intellectual contributions throughout the project. All authors reviewed the manuscript, approved the final version to be published, and agreed to be accountable for all aspects of this study.

Disclosures: All authors have completed and submitted the ICMJE Form for Disclosure of Potential Conflicts of Interest. T. Kato reported personal fees from "Daiichi Sankyo Co., Ltd." during the conduct of the study; in addition, T. Kato had a patent to WO2020179859 A1 pending "Daiichi Sankyo Co., Ltd." Y. Yoshihama reported personal fees from "Daiichi Sankyo Co., Ltd." during the conduct of the study; in addition, Y. Yoshihama had a patent to WO2020179859 A1 pending "Daiichi Sankyo Co., Ltd." No other disclosures were reported.

Submitted: 22 December 2022

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

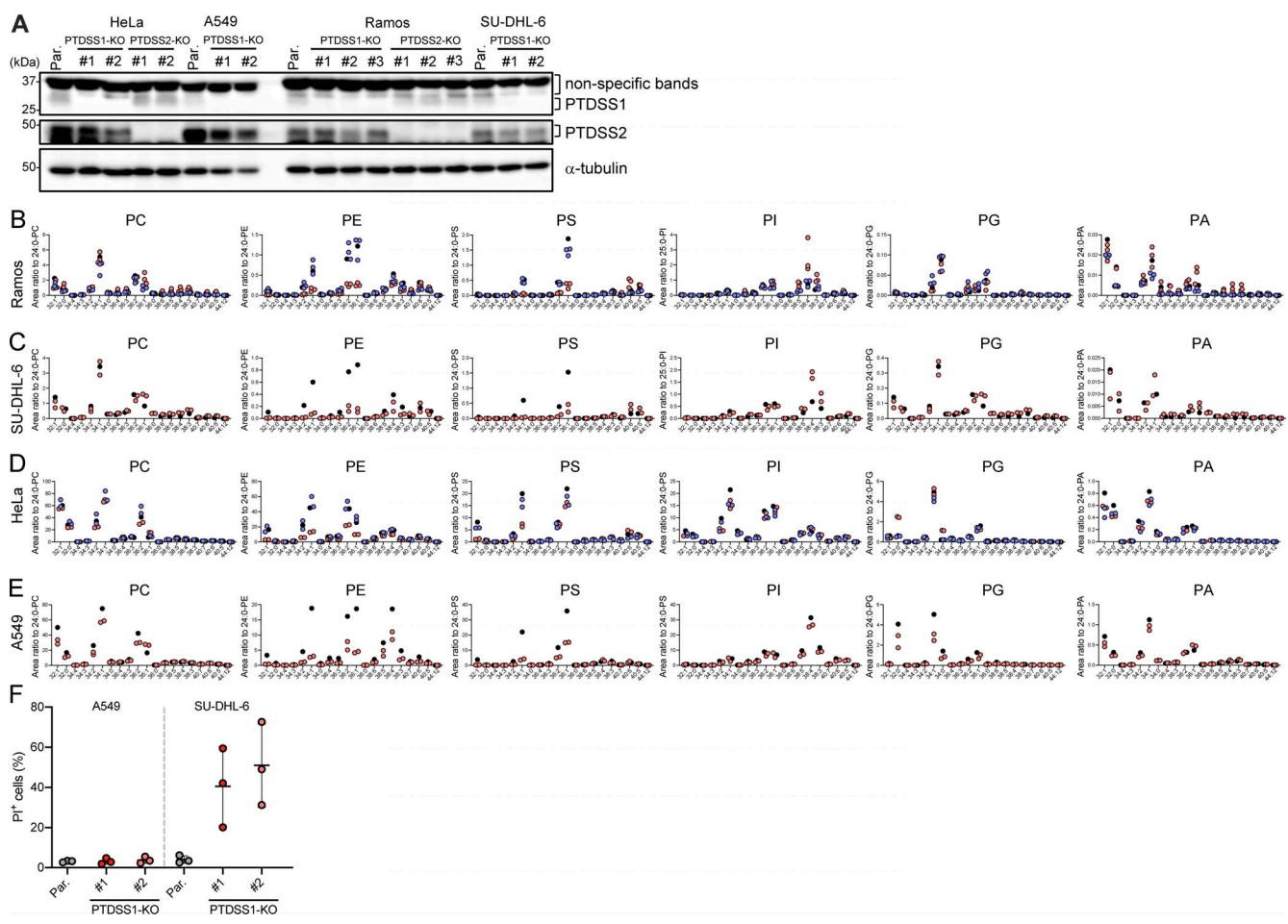

Figure S1. **Gene knockout of PTDSS1 significantly alters the phospholipidome of lymphoma cells. (A)** The expression of PTDSS1 and PTDSS2 in HeLa, A549, Ramos, and SU-DHL-6 cells. The specific bands for PTDSS1 or PTDSS2 are indicated. Non-specific bands appeared above the correct bands. **(B–E)** The level of phospholipid species in Ramos (B), SU-DHL-6 (C), HeLa (D), and A549 (E) clones. Data are presented as an area ratio to the values of internal standard (1 μM), and each dot indicates each clone (red and blue for PTDSS1-KO and PTDSS2-KO, respectively), or parental cells (black). **(F)** The PI[+] dead cell accumulation of A549 or SU-DHL-6 clones during the culture. The clones were stained with PI for 10 min, and then PI[+] cells were detected using flowcytometry. Data are presented as mean ± SD of three independent experiments with a dot plot of values for the individual experiments.

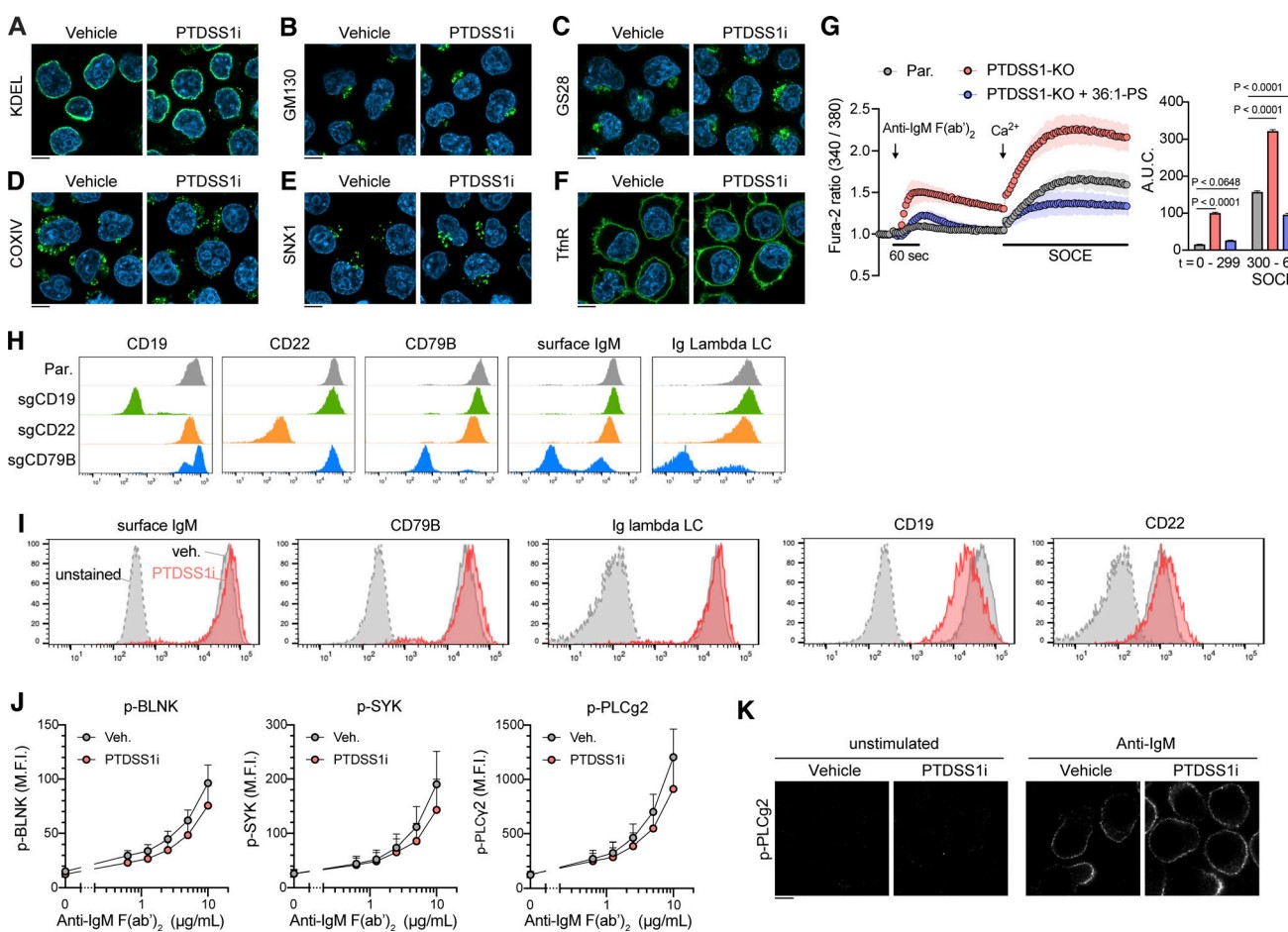

Figure S2. **PTDSS1 inhibition did not overtly affect the morphology of organelles, cell surface expression of BCR signaling components, or BCR-induced phosphorylation of downstream molecules. (A–E)** Ramos cells were cultured for 3 d with 100 nM PTDSS1i and then fixed with PFA. The intra-cellular localization of each organelle-resident protein was analyzed by immunocytochemical staining using specific antibodies against KDEL (A) for ER, GM130 (B) and G28 (C) for Golgi, COXIV (D) for mitochondria, and SNX1 (E) for endosomes. **(F)** The PM was visualized by staining using an anti-Transferrin receptor (TfnR) antibody. The fluorescent images were analyzed using confocal microscopy. Scale bar represents 5 µm. **(G)** SOCE responses of Ramos cells to BCR ligation-induced store $Ca^{2+}$ depletion. Fura-2-stained Ramos cells were stimulated with 20 µg/ml anti-IgM F(ab')$_2$, and then 2 mM $Ca^{2+}$ was added to evaluate the $Ca^{2+}$ efflux (left panel). The values for area under curve (A.U.C.) within the indicated periods are shown (right panel). Data are presented as the mean ± SD of three independent experiments. Statistical P values examined based on Bonferroni's multiple comparison tests are shown. **(H)** The cell surface expression of CD79B, CD19, CD22, and IgM on bulk Ramos cells transfected with pSpCas9(BB)-2A-GFP (PX458) vector containing each sgRNA sequence. The GFP-positive Ramos cells enriched with FACS were stained with the fluorescence-conjugated primary antibodies against each molecule. The negative populations for the cell surface antigen (typically ~50% present in total cells) were further enriched and used as a bulk KO line. **(I)** The cell surface expression of BCR components. Ramos cells cultured with 100 nM PTDSS1i for 3 d were stained with the fluorescence-conjugated antibodies, and then analyzed by flowcytometry. **(J)** Ramos cells cultured as described in A were stimulated with the indicated concentrations of anti-IgM F(ab')$_2$ at 37°C for 4 min, and then fixed with PFA-based fixative, stained with the fluorescence-conjugated antibodies, and then analyzed by flowcytometry. Data are presented as median fluorescent intensity (M.F.I.) values (mean + SD of three independent experiments). **(K)** The effect of PTDSS1 inhibition on subcellular localization of phosphorylated PLCγ2. Ramos cells stained as described in J were analyzed using confocal microscopy. Scale bar represents 5 µm. Source data are available for this figure: SourceData FS2.

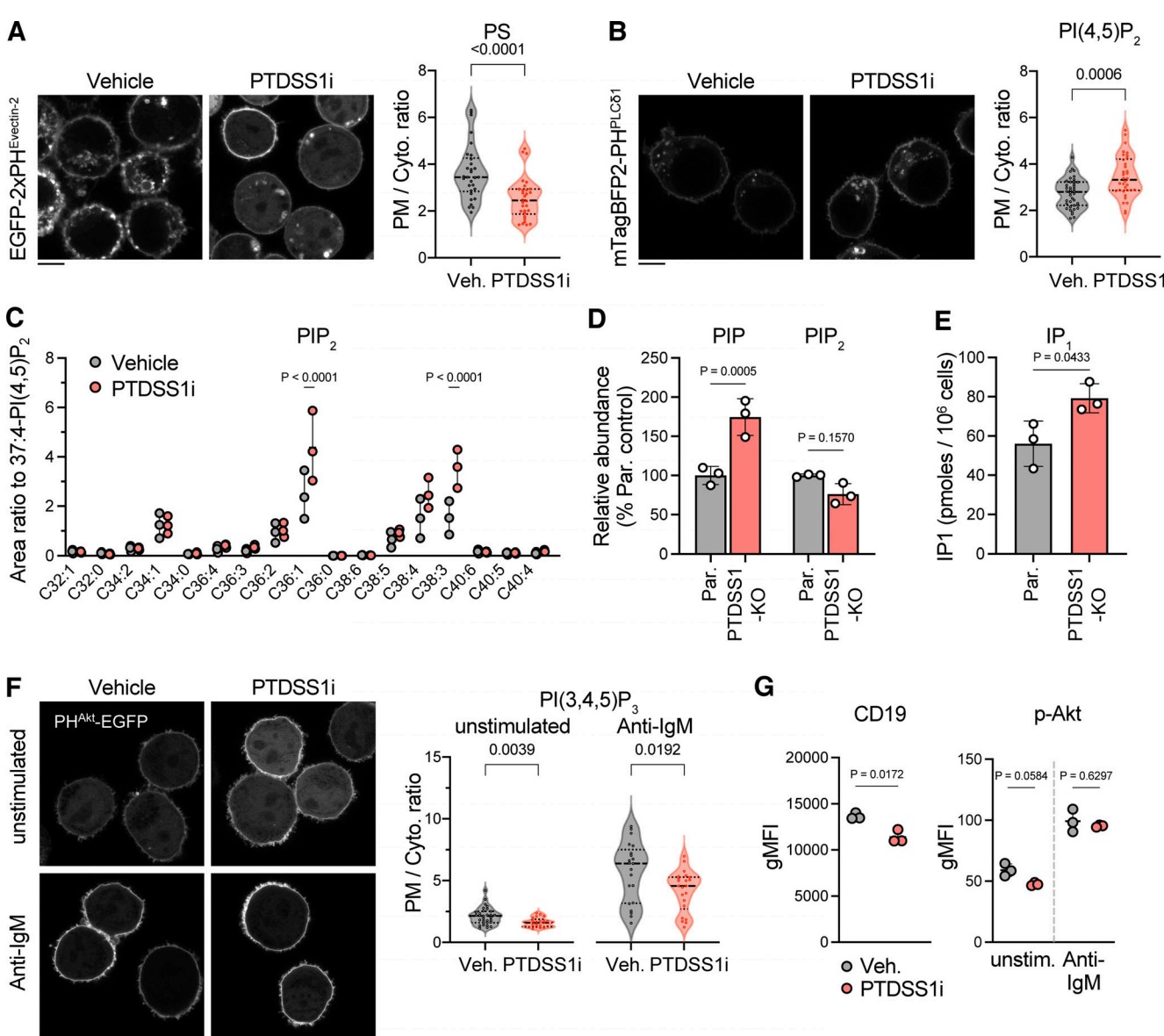

Figure S3.    **The effects of PTDSS1 inhibition on the PIPs levels and PIPs-related signaling. (A and B)** Intracellular distribution of PS or PI(4,5)P$_2$ in vehicle- or PTDSS1i-treated Ramos cells. Ramos cells stably expressing EGFP-2xPH$^{Evectin2}$ (A) or mTagBFP2-2xPH$^{PLCδ1}$ (B) were cultured for 24 h with 100 nM PTDSS1i and then fluorescent images were analyzed using confocal microscopy (left). Fluorescence intensity ratio of PM to cytosol (PM/Cyto.) was quantified and dot-plotted for each cell (right). Statistical P values based on Mann–Whitney $U$ test are shown. The scale bar represents 5 μm. **(C)** PIP$_2$ levels in vehicle- or PTDSS1i-treated Ramos cells. Data are presented as the mean ± SD of three independent experiments with a dot plot of values for the individual experiments. Statistical P values based on one-way ANOVA followed by Šídák's multiple comparisons test are shown. **(D)** PIP and PIP$_2$ levels in PTDSS1-KO Ramos cells. Data are presented as the mean ± SD of three independent cultures. Statistical P values based on one-way ANOVA followed by Šídák's multiple comparisons test are shown. **(E)** Enhanced IP$_1$ production in PTDSS1-KO Ramos cells. The cells were stimulated with 20 μg/ml anti-IgM F(ab')$_2$ in the presence of LiCl. Data are presented as the mean ± SD of three independent cultures. Statistical P values based on $t$ test are shown. **(F)** Intracellular distribution of PI(3,4,5)P$_3$ in vehicle- or PTDSS1i-treated Ramos cells. Ramos cells stably expressing PH$^{Akt}$-EGFP cultured for 24 h with 100 nM PTDSS1i were stimulated with 20 μg/ml anti-IgM F(ab')$_2$ for 5 min and then fluorescent images were analyzed using confocal microscopy (left). Fluorescence intensity ratio of PM to cytosol (PM/Cyto.) was quantified and dot-plotted for each cell (right). Statistical P values based on Mann–Whitney $U$ test are shown. Scale bar represents 5 μm. **(G)** The effects of PTDSS1 inhibition on the cell surface expression of CD19 or phosphorylation status of Akt. Ramos cells cultured with 100 nM PTDSS1i for 3 d were stained with the fluorescence-conjugated anti-CD19 antibody (left) or intracellularly labeled with the fluorescence-conjugated anti-phosphorylated Akt (p-Akt) antibody, and then analyzed by flowcytometry. Data are presented as geometric mean fluorescent intensity (gMFI) values (mean ± SD of three independent cultures). Statistical P values based on one-way ANOVA followed by Šídák's multiple comparisons test are shown.

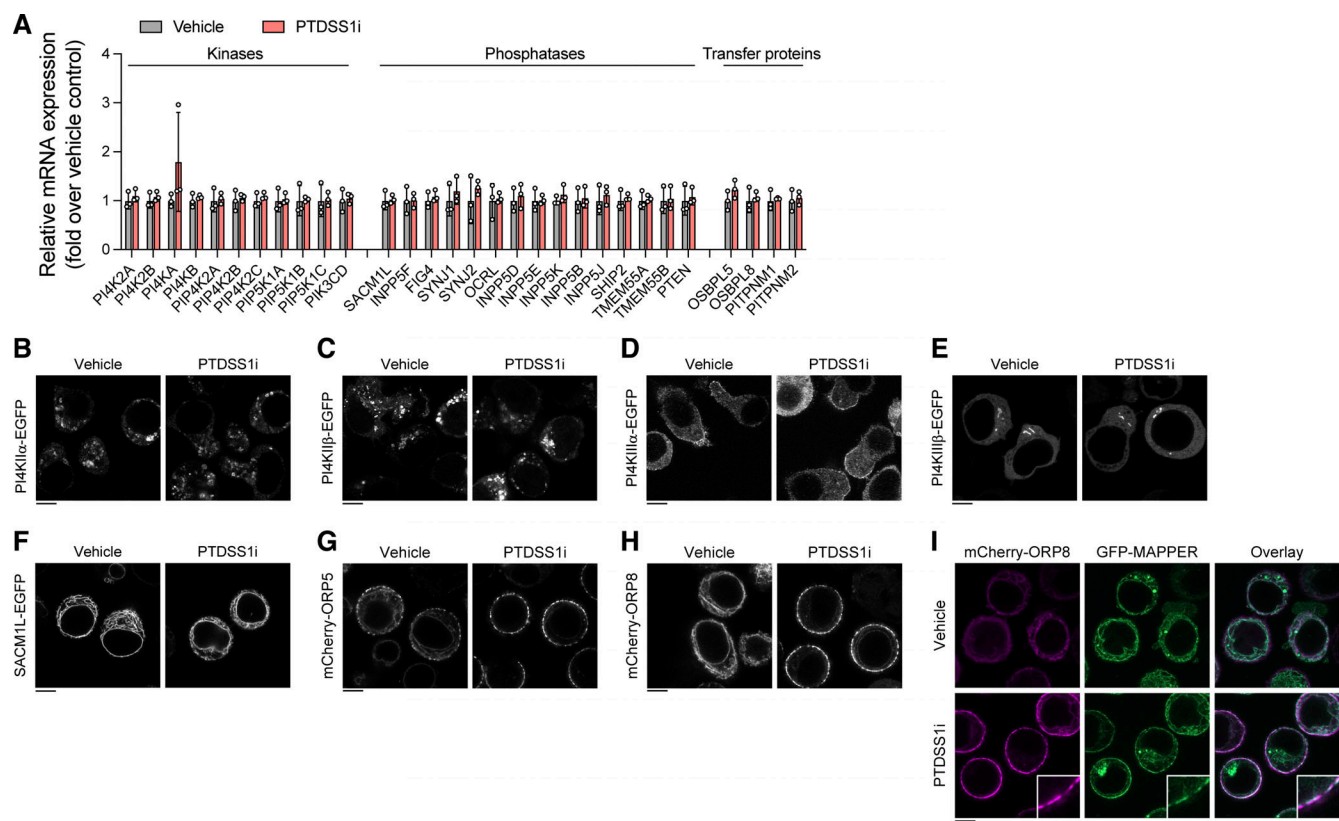

Figure S4. **The effects of PTDSS1 inhibition on expression and subcellular localization of kinases, phosphatases, and transfer proteins for phosphoinositides. (A)** The effects of PTDSS1 inhibition on mRNA expression of various phosphoinositide kinases, phosphatases, and transfer proteins. Total RNA collected from Ramos cells cultured with 100 nM PTDSS1i for 3 d were subjected to qRT-PCR analysis. Data are presented as mean ± SD of three independent cultures. **(B–H)** Intracellular localization of phosphoinositide kinases, phosphatases, and transfer proteins in vehicle- or PTDSS1i-treated Ramos cells. Ramos cells stably expressing PI4KIIα-EGFP (B), PI4KIIβ-EGFP (C), PI4KIIIα-EGFP (D), PI4KIIIβ-EGFP (E), SACM1L-EGFP (F), mCherry-ORP5 (G), or mCherry-ORP8 (H) were cultured for 24 h with 100 nM PTDSS1i and then fluorescent images were analyzed using confocal microscopy. Scale bar represents 5 μm. **(I)** Colocalization of mCherry-ORP8 with GFP-MAPPER. Ramos cells stably expressing mCherry-ORP8 and GFP-MAPPER were cultured for 24 h with 100 nM PTDSS1i and then fluorescent images were analyzed using confocal microscopy. Scale bar represents 5 μm. Insets show magnified fields.

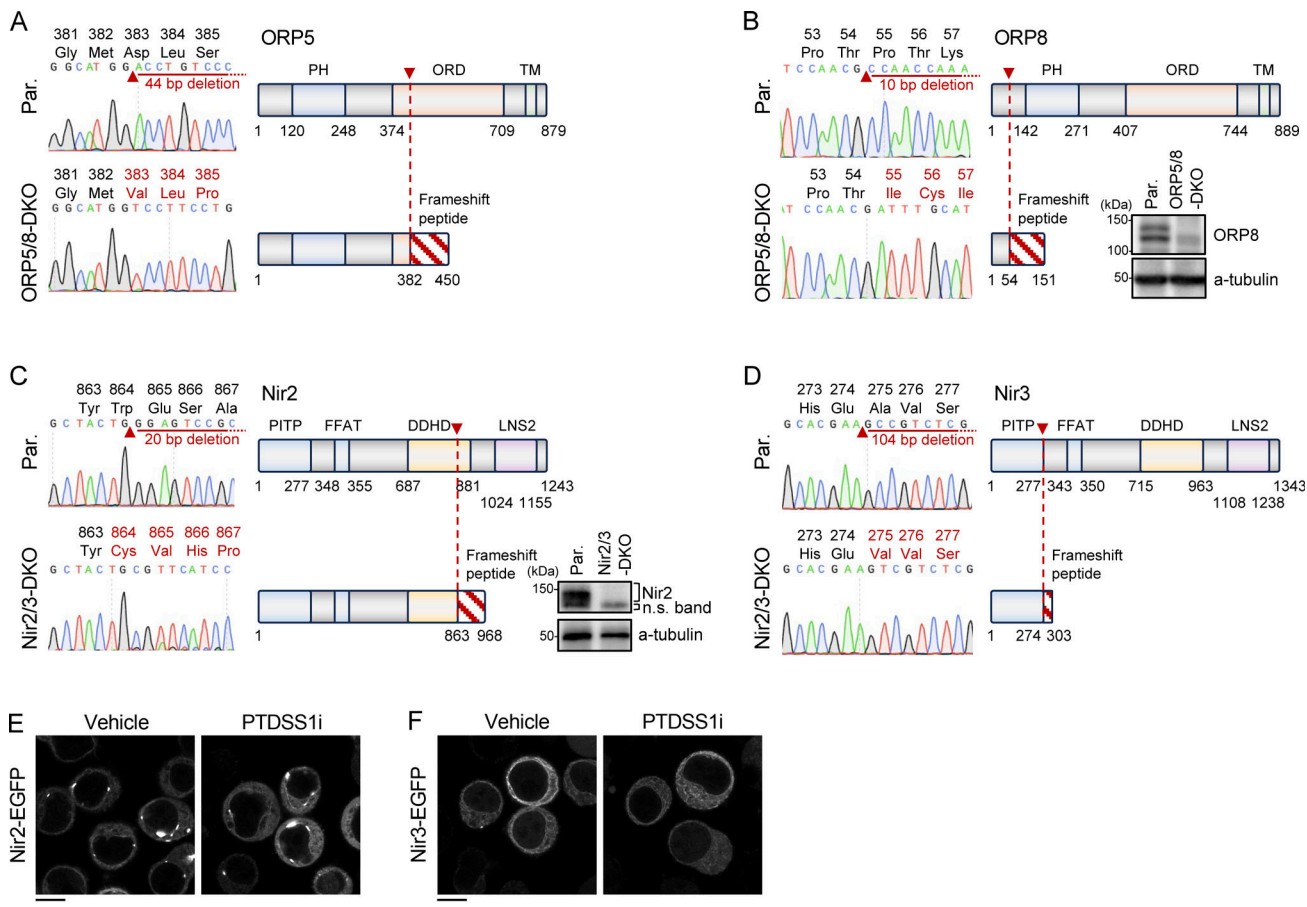

Figure S5. **Generation of ORP5/8-DKO and Nir2/3-DKO Ramos cells. (A–D)** Confirmation of CRISPR-Cas9-mediated genome editing for ORP5 (A), ORP8 (B), Nir2 (C), and Nir3 (D). The left panel in each figure shows the electropherogram for the direct genomic sequence with corresponding amino acids sequence around the indel mutation site indicated by a red arrowhead, and the right panel in each figure shows a schematic of the domain structure for each molecule. All those deletions cause a frameshift peptide with an early stop codon indicated by a diagonal line, thus leading to the KO of functional protein. For ORP8 (B) and Nir2 (C), protein KO was also confirmed by Western blot. **(E and F)** The effects of PTDSS1 inhibition on subcellular localization of Nir2 and Nir3. Ramos cells stably expressing Nir2-EGFP (E) or Nir3-EGFP (F) were cultured for 24 h with 100 nM PTDSS1i and then fluorescent images were analyzed using confocal microscopy. Scale bar represents 5 μm. Source data are available for this figure: SourceData FS5

**Provided online are Table S1 and Table S2. Table S1 lists human cancer cell lines used in this study. Table S2 lists oligonucleotides used in this study.**

