## [Peer Review File · The Journal of Cell Biology]

Phosphatidylserine Synthesis Controls Oncogenic B Cell Receptor Signaling in B Cell Lymphoma

Jumpei Omi, Taiga Kato, Yohei Yoshihama, Koki Sawada, Nozomu Kono, and Junken Aoki

Corresponding Author(s): Junken Aoki, The University of Tokyo

Review Timeline:

Submission Date:	2022-12-22
Editorial Decision:	2023-02-21
Revision Received:	2023-09-13
Editorial Decision:	2023-10-27
Revision Received:	2023-11-04

Monitoring Editor: William Prinz

Scientific Editor: Tim Fessenden

Transaction Report:

DOI: <https://doi.org/10.1083/jcb.202212074>

February 21, 2023

Re: JCB manuscript #202212074

Prof. Junken Aoki
The University of Tokyo
Department of Health Chemistry, Graduate School of Pharmaceutical Sciences
7-3-1, Hongo
Bunkyo-ku, Tokyo 113-0033
Japan

Dear Prof. Aoki,

Thank you for submitting your manuscript entitled "Phosphatidylserine Synthesis Controls Oncogenic B Cell Receptor Signaling in B Cell Lymphoma". After a longer than usual delay, for which we sincerely apologize, your manuscript has been evaluated by expert reviewers whose reports are appended below. Unfortunately, after an assessment of the reviewer feedback, our editorial decision is against publication in JCB at this time.

You will see that the views are mixed. Reviewer 2 was enthusiastic but raises some issue that should be addressed. The other two reviewers have more substantial, related concerns. Most importantly, both feel you need more mechanistic insight into the regulation of PIPs and calcium levels when PTDSS1 is inhibited or knocked down (Rev 1, points 2,3 and Rev 3, 2nd paragraph). Reviewer 3 also raises important concerns about the efficiency of knock down of ORP5/ORP8 and whether changes in PI levels after reducing PTDSS1 activity are mostly mediated by loss of ORP5/8 function (2nd paragraph). The other concerns of reviewers 1 and 3 should be addressed, though it is less important to determine the subcellular distribution PIPs and other lipids (Rev 1, pt 1).

Although your manuscript is intriguing, I feel that the points raised by the reviewers are more substantial than can be addressed in a typical revision period. If you wish to expedite publication of the current data, it may be best to pursue publication at another journal.

Given our interest in the topic, I would be open to resubmission to JCB of a significantly revised and extended manuscript that fully addresses the reviewers' concerns and is subject to further peer-review. If you would like to resubmit this work to JCB, please contact the journal office to discuss an appeal of this decision or you may submit an appeal directly through our manuscript submission system.

Regardless of how you choose to proceed, we hope that the comments below will prove constructive as your work progresses. We would be happy to discuss the reviewer comments further once you've had a chance to consider the points raised in this letter. You can contact the journal office with any questions, cellbio@rockefeller.edu or call (212) 327-8588.

Thank you for thinking of JCB as an appropriate place to publish your work.

Sincerely,

William Prinz
Monitoring Editor
Journal of Cell Biology

Tim Fessenden
Scientific Editor
Journal of Cell Biology

Reviewer #1 (Comments to the Authors (Required)):

In this manuscript, Omi et al have performed a screening of cancer cell lines with a PTDSS1 inhibitor (PTDSS1i) and found that B lymphoma cell lines highly depend on PS synthesis for their survival. Pharmacological inhibition by PTDSS1i as well as genetic ablation of PTDSS1 leads to phospholipid imbalance, growth suppression and apoptotic cell death in Ramos cells. Mouse xenograft experiments also reveal the vulnerability of B cell lymphoma to PTDSS1 inhibition in vivo. Mechanistically,

PTDSS1 KO cells show an enhanced cytosolic Ca²⁺ elevation and SOCE, and are more apoptotic. PLCg2 inhibitor or genetic ablation of genes involved in BCR signaling such as CD79B, PLCg2 or CD19 attenuates the efficacy of PTDSS1i. BCR stimulation results in sustained higher PI4P and possibly PIP2, and more IP1 production in PTDSS1i-treated cells. Finally, the authors demonstrate similar defects in cellular phosphoinositide levels and Ca²⁺ mobilization property in ORP5/8 KD Ramos cells. Based on these results, the authors conclude that PTDSS1 and ORP5/8 control the levels of PIPs in the PM and magnitude of Ca²⁺ response upon BCR stimulation.

This is a well written manuscript that describes potentially interesting observations. PS dependency, lipid-biochemical property and Ca²⁺ response of B cell lymphoma treated with PTDSS1i or deficient in PTDSS1 gene are well characterized. However, cell biological evidence for the mechanism underlying the regulation of BCR signaling by PTDSS1-mediated PS synthesis is rather weak and incomplete. Thus, I can't support the publication of the current manuscript at JCB. I would encourage the authors to consider the following points, which would strengthen the current manuscript.

Major points:

1. Subcellular distribution of lipids, especially PS and PIPs such as PI4P, PIP2 and PIP3, should be investigated. Are PI4P and PIP2 indeed higher at the PM in PTDSS1 KO cells as the authors speculated? Do they change their distribution during BCR stimulation? How about PS distribution/behavior in KO?
2. How do PI, PIP and PIP2 increase in PTDSS1 KO or PTDSS1i-treated cells? There may be a number of possibilities: Do their kinases/phosphatases up/down-regulated? How about their localization? Alternatively, lipid transport regulation might be affected. Expression or localization of ORP5/8 changed in PTDSS1 KO cells? How about Nir2/3, PI/PA transfer proteins? I would expect mechanistic evidence for the regulation of PIPs by PTDSS1.
3. What is the mechanism behind the elevated production of IP1 in PTDSS1 KO cells? Do the authors propose that this is due to higher levels of PIP2 in PTDSS1 KO? If so, are the localization and behavior of PLCg2 changed in PTDSS1 KO cells with/without BCR stimulation? Additionally, can the authors test if reduction of PIP2 and/or PIP3 (by overexpression of PIP2 or PIP3 phosphatases or some other acute recruitment techniques) rescues the elevated production of IP1 and Ca²⁺ response in PTDSS1 KO cells? If the authors propose different mechanism(s), please provide such evidence.
4. Are organelles such as mitochondria, endosomes, lysosomes, Golgi and the ER, and the PM in PTDSS1 KO and PTDSS1i-treated cells not overtly affected in their number, position and morphology with/without BCR stimulation?

Minor points:

1. As a control for rescue experiments by adding PS in the culture media, irrelevant lipids such as PC should be tested.
2. Please indicate the statistical values for Fig. 5G, 6D and 6E. Are they statistically significant?

Reviewer #2 (Comments to the Authors (Required)):

The manuscript reports an interesting effect of PTDSS1 targeting on the survival of lymphoma cells. Instead of synthetic lethality with PTDSS2, the lymphoma cells are sensitive in part through enhanced BCR-mediated apoptosis. This indicates that PS synthesis and balanced levels of PI and PE are important selectively in these haematopoietic cancers. The results are novel and I find the experiments interesting, informative and mostly well done. The conclusions about how PS synthesis regulates oncogenic (and tumour-suppressive) BCR signalling should be important for the field. I do have some questions, however, about the mechanisms by which the phospholipid imbalance causes apoptosis.

Major issues:

1. The experiments on BCR-induced apoptosis are very interesting and so is the idea that enhanced BCR calcium signalling is responsible for the apoptosis of the PTDSS1i-treated cells. I don't think any previous studies actually showed spontaneous calcium oscillations in lymphoma studies, so this is novel. Implicating calcium signalling in apoptosis is reasonable (see e.g. Akkaya Nat Immune 2018), however, there are no experiments that directly support this conclusion (as opposed to another signalling pathway downstream of the BCR causing the apoptosis). Can the authors selectively block calcium signalling (e.g. inhibition or KO of Orai etc.) to see if this rescues the apoptosis mediated by PTDSS1i?
2. The rescue experiments, which knock out BCR signalling proteins or use IgM-negative Ramos cells are critical, but need some clarification. First, chronic oncogenic BCR signalling has been reserved for ABC-type DLBCL and is associated with BTK-PLCγ2-NFκB signalling. In Burkitt lymphoma cells (Ramos) and GCB-type DLBCL (SUDHL6), the oncogenic effects are driven by the "tonic" CD19-PI3K pathway. The cited papers actually refer to a range of leukaemia and lymphoma studies that describe a variety of mechanisms. So it would be clearer not to call the BCR signalling "chronic". Second, it is expected that KO of CD19 or CD79B will retard the growth of the Ramos cells. Thus, the experiment in Fig 5G is problematic as the data has been normalised, making it impossible to see this. Without taking this baseline effect into account, it is difficult to judge the rescue

effect on the PTDSS1i-treated cells. The KO of PLC γ 2 is a better experiment, because PLC γ 2 is not expected to be essential for the growth of Ramos cells, but does regulate calcium signalling. Unfortunately, the PLC γ 2 KO cells have been obtained by selection involving resistance to apoptosis, making it unclear whether it is PLC γ 2 or the selection that is responsible for the effect. Ideally, PLC γ 2 KO could be done with an expression marker that could allow monitoring the growth of the KO cells compared to WT without this selection. Similarly, the experiment with the Ramos cells that spontaneously lost IgM is problematic, because we do not know how these cells adapted to their BCR loss. Finally, I understand from the discussion that the authors are aware that the rescue here is not complete and thus other BCR-independent mechanisms are at play (PE?). But it may be fair to bring this point here as well.

3. The authors do not take into account PIP3 levels and phosphorylation of Akt. This is a critical mechanisms required for the growth of Ramos cells. Are PIP3 levels and Akt phosphorylation affected by PTDSS1 inhibition or knock out?

Minor comments:

4. PTDSS1 WB shows multiple bands. The authors may want to illustrate their confidence that this is the correct band and highlight it better.

5. Line 124 is unclear. The A549 cells are not affected, but the SUDHL6 are.

6. Fig 3b the red line for the PTSS1 KO is invisible, (it should overlap with 0?).

7. Fig 5 - the numbers of spikes should be converted to frequency so that they are independent of the duration of data collection.

8. Why is the KO of ORP5+8 increasing the total levels of PIPs? According to Fig 7, this will affect translocation. But as far as I can see the lipid measurements are from the whole cell, not just the PM. Does the translocation, lead to destruction? Please explain.

9. I suggest that supplementary Fig 5C,D described in "Results" to help to explain that lymphoma also likely suffer from reduced PE. Since the focus is on the PIP2-IP3-Calcium signalling, this is fine to leave open, but it is important to acknowledge in the main part of the paper.

Reviewer #3 (Comments to the Authors (Required)):

In this manuscript by Omi et al., the authors showed that inhibition or depletion of PTDSS1, a key enzyme for phosphatidylserine (PS) synthesis, results in increased phosphatidylinositol (PI) lipids, cytosolic calcium signals, and cell death in B cell receptor (BCR)-positive lymphoma cell lines. The authors suggest that targeting PS synthesis may be a new strategy to treat B cell lymphoma.

This manuscript presents a lot of results. Nevertheless, there is a lack of new mechanistic insights into the elevated PI lipids and calcium signaling caused by PTDSS1 deficiency. The authors proposed a model that aberrant PIP/calcium levels and apoptosis are caused by abolished PS/PIP exchange at ER-PM contact sites. However, the effects of PTDSS1 deficiency on PI lipids are much bigger than knockdown of both ORP5 and ORP8 that mediate PS/PIP exchange at ER-PM contact sites (note: the differences in the scale of the Y-axis of Figures 6B/6C vs Figures 6F/6G). Also, it's unclear whether ORP5/ORP8 expression is significantly reduced in ORP5/ORP8 double knockdown Ramos Cells in these experiments. The results do not provide sufficient support for the proposed model.

It is interesting to note that PA is elevated, like PI lipids, in most PTDSS1i-treated cancer cells (Figure 1). Elevated PA has been shown to trigger targeting of PI/PA transfer proteins Nir2/Nir3 to ER-PM contact sites for replenishing plasma membrane PI(4,5)P2 following receptor-induced PLC activation (PMID: 25887399). Nevertheless, the elevation of PA in PTDSS1 inhibited cells was not discussed in this study.

Hyper-responsiveness to T cell receptor signaling has been shown in PTEN-deficient Jurkat T cells (PMID: 10958690). As shown in Figure 1, Jurkat cells exhibited higher PI lipids and cell death, like B lymphoma cell lines used in most figures of this manuscript, following PTDSS1 inhibitor treatment. Is elevated BCR signaling caused by PTEN deficiency in B lymphoma lines used in this study? If so, PTEN should be incorporated into the proposed model of this manuscript.

September 13, 2023

Dr. Tim Fessenden
Scientific Editor
Journal of Cell Biology

Dear Dr. Tim Fessenden,

Thank you for considering our manuscript entitled "Phosphatidylserine Synthesis Controls Oncogenic B Cell Receptor Signaling in B Cell Lymphoma," (JCB manuscript #202212074) for publication in *Journal of Cell Biology*.

In this manuscript, we revealed that PS synthesis by PTDSS1 functions as an important regulator of oncogenic BCR signal transduction in B cell lymphoma. We believe that the revised manuscript warrants further consideration in *Journal of Cell Biology*, because the novelty and importance of the concept have been substantially strengthened by new experiments and data that address the reviewer's prior concerns.

The reviewers raised two major concerns regarding 1) unclearness about the mechanism of how the suppressed function of PTDSS1 lead to enhanced Ca²⁺ signaling and increased PIPs level (Reviewers 1 and 3) and 2) shortness of data showing the involvement of Ca²⁺ signaling and PIP₃-Akt signaling in PTDSS1i-induced apoptosis (Reviewer 2). In answer to the first concern, we generated ORP5/8-double knockout (DKO) Ramos cells to avoid the ambiguity of the knockdown experiment in the original manuscript, which reviewer 3 pointed out. We found again that ORP5/8-DKO only partially phenocopied the effects of PTDSS1 inhibition (**Fig. 7, C-E**), suggesting that the additional mechanism responsible for PIPs regulation by PTDSS1. Therefore, following the suggestions by reviewers 1 and 3, we additionally generated Nir2/3-DKO cells and analyzed their PIP levels. We found that Nir2/3-DKO significantly impaired the elevation of both PI and PIP levels upon PTDSS1 inhibition (**Fig. 7, C-E and G**). Based on these new results, we propose the revised model, where both

the deceleration of PS/PI4P countertransport by ORP5/8 and acceleration of PI-cycle by Nir2/3 are the molecular mechanisms underlying the elevation of phosphoinositides and Ca²⁺ levels upon PTDSS1 inhibition (**Fig. 8**).

In answer to the second concern, we performed additional experiments using several inhibitors and demonstrated that pharmacological inhibition of both Ca²⁺ release via IP₃ receptors and Ca²⁺ entry via the Orai channel significantly rescued PTDSS1i-induced apoptosis (**Fig. 5H**). We found that PTDSS1 inhibition slightly reduced the PM PIP₃ levels, but it did not lead to a significant decrease in phosphorylation of Akt (**Supplementary Fig. 6F-G**). These new data clearly demonstrate that enhanced Ca²⁺ signaling is responsible for PTDSS1i-induced apoptosis in B cell lymphoma.

In the revised manuscript, we have addressed all the other concerns of the reviewers, including the effects of PTDSS1 inhibition on the expression and localization of PI kinases and phosphatases and the subcellular distribution of PS and PIPs. Accordingly, we have added or renewed Figs. 4B, 4C, 5G-H, 6C-F, 7A-G, 8, Supplementary Figs. 3, 4, 5C, 6, 7, 8, 9, and 10 in the revised manuscript, and deleted Figs. 5H, 6D-H, Supplementary Figs. 3B-C, and 5C-D in the original manuscript. All changes in the text of the revised manuscript are highlighted in yellow.

Our point-to-point responses to the reviewer's comments are also provided on the pages that follows.

We greatly appreciate your assistance in the review of our manuscript and look forward to hearing from you.

Sincerely,

Junken Aoki, Ph.D.

Graduate School of Pharmaceutical Sciences, The University of Tokyo,

7-3-1 Hongo, Bunkyo-ku, Tokyo 113-86514, Japan

Tel and Fax: +81-358-41-4736

e-mail: jaoki@mol.f.u-tokyo.ac.jp

Point-by-point responses to the reviewers' comments

Reviewer #1 (Comments to the Authors (Required)):

In this manuscript, Omi et al have performed a screening of cancer cell lines with a PTDSS1 inhibitor (PTDSS1i) and found that B lymphoma cell lines highly depend on PS synthesis for their survival. Pharmacological inhibition by PTDSS1i as well as genetic ablation of PTDSS1 leads to phospholipid imbalance, growth suppression and apoptotic cell death in Ramos cells. Mouse xenograft experiments also reveal the vulnerability of B cell lymphoma to PTDSS1 inhibition in vivo. Mechanistically, PTDSS1 KO cells show an enhanced cytosolic Ca²⁺ elevation and SOCE, and are more apoptotic. PLC β 2 inhibitor or genetic ablation of genes involved in BCR signaling such as CD79B, PLC β 2 or CD19 attenuates the efficacy of PTDSS1i. BCR stimulation results in sustained higher PI4P and possibly PIP₂, and more IP₁ production in PTDSS1i-treated cells. Finally, the authors demonstrate similar defects in cellular phosphoinositide levels and Ca²⁺ mobilization property in ORP5/8 KD Ramos cells. Based on these results, the authors conclude that PTDSS1 and ORP5/8 control the levels of PIPs in the PM and magnitude of Ca²⁺ response upon BCR stimulation.

This is a well written manuscript that describes potentially interesting observations. PS dependency, lipid-biochemical property and Ca²⁺ response of B cell lymphoma treated with PTDSS1i or deficient in PTDSS1 gene are well characterized. However, cell biological evidence for the mechanism underlying the regulation of BCR signaling by PTDSS1-mediated PS synthesis is rather weak and incomplete. Thus, I can't support the publication of the current manuscript at JCB. I would encourage the authors to consider the following points, which would strengthen the current manuscript

Response: We would like to thank the reviewer #1 for reviewing our manuscript and offering constructive suggestions that help strengthen our conclusion. We have addressed all of the reviewer's concerns in the point-by-point responses as follows, especially with more clear data about the mechanistic evidence for the regulation of PIPs and IP₁ by PTDSS1 as

described in the response to the major point #2.

Major points:

1. Subcellular distribution of lipids, especially PS and PIPs such as PI4P, PIP2 and PIP3, should be investigated. Are PI4P and PIP2 indeed higher at the PM in PTDSS1 KO cells as the authors speculated? Do they change their distribution during BCR stimulation? How about PS distribution/behavior in KO?

Response 1: Following the reviewer's suggestion, we examined the subcellular distribution of phospholipids by generating Ramos cells stably expressing EGFP-tagged specific probes, such as 2xPH domain of Eevctin2 for PS, 2xP4M of SidM for PI4P, PH domain of PLCd1 for PI(4,5)P₂, and PH domain of Akt1 for PI(3,4,5)P₃. We performed the experiments by using the PTDSS1 inhibitor (PTDSS1i), but not PTDSS1-KO clones, to avoid the unfavorable variation of expression efficiency of these ectopically transduced probes between parent and KO-clones. We found that the probes for PI4P and PI(4,5)P₂ showed a higher PM/Cyto ratio in PTDSS1i-treated cells (**Fig. 6C** and **Supplementary Fig. 6B**). By contrast, PTDSS1i treatment resulted in a substantial decrease in PM staining by the probe for PS (**Fig. 6C** and **Supplementary Fig. 6A**). These observations clearly demonstrated that PI4P and PI(4,5)P₂ are indeed higher while PS is lower at the PM in PTDSS1i-treated cells, which, we believe, strengthens our working model.

Unlike PI4P and PI(4,5)P₂, we found a lower PM/Cyto ratio for PI(3,4,5)P₃ probe in PTDSS1i-treated cells, even under the BCR ligation condition that significantly increased the PM/Cyto ratio of PI(3,4,5)P₃ probe (**Supplementary Fig. 6F**). These observations were consistent with the downregulation of CD19 (**Supplementary Fig. 6G left**), which promotes PI(3,4,5)P₃ production via PI3K activation. PTDSS1 inhibition slightly but not significantly reduced the phosphorylation of Akt under the unstimulated condition (**Supplementary Fig. 6G right**). Therefore, we concluded that the decrease in the PI(3,4,5)P₃ level at the PM did not contribute to the survivability of Ramos cells under the PTDSS1i inhibition. We did not observe overt changes in PI4P and PI(4,5)P₂ distribution during BCR stimulation. We added Fig. 6C and Supplementary Fig. 6A, 6B, 6F, and 6G, and accordingly changed the Results

(lines 210-215 and 228-234) in the revised manuscript.

2. How do PI, PIP and PIP₂ increase in PTDSS1 KO or PTDSS1i-treated cells? There may be a number of possibilities: Do their kinases/phosphatases up/down-regulated? How about their localization? Alternatively, lipid transport regulation might be affected. Expression or localization of ORP5/8 changed in PTDSS1 KO cells? How about Nir2/3, PI/PA transfer proteins? I would expect mechanistic evidence for the regulation of PIPs by PTDSS1.

Response 2: As the reviewer pointed out, the additional mechanisms other than ORP5/8 may also contribute to the PI and PIPs elevation in PTDSS1i-treated cells because the effects of PTDSS1 inhibition on PIPs levels were greater than ORP5/8-double knockdown (**Fig. 6B, C vs 6F, G** in the original manuscript). We addressed the possibilities point-by-point below (**Response 2-1, 2-2, and 2-3**).

2-1. PTDSS1 inhibition did not affect mRNA expression or subcellular localization of PI kinases and PIPs phosphatases. Considering that the changes in PIP level were much greater than those in PIP₂ (**Fig. 6B vs Supplementary Fig. 6C** in the revised manuscript), PTDSS1 inhibition may primarily affect PIP levels rather than PIP₂ itself. Therefore, following the reviewer's suggestion, we examined the mRNA expression levels (**Supplementary Fig. 7A**) and localization (**Supplementary Fig. 7B-7F**) of PI kinases and PIP phosphatase. We found that PTDSS1 inhibition did not cause significant changes in mRNA levels of 12 PI/PIPs kinases and 14 PIPs phosphatases as judged by qRT-PCR analysis (**Supplementary Fig. 7A**). Similarly, we did not observe overt effects of PTDSS1 inhibition on subcellular localization of ectopically expressed PI kinases producing PI4P (PI4KIIa, PI4KIIb, PI4KIIIa, and PI4KIIIb) and phosphatase for PI4P (SACM1L).

2-2. PTDSS1 inhibition induces the translocation of ORP5/8 to the PM-ER contact sites. Insufficient PS synthesis causes deceleration of PS/PI4P countertransport between the ER-PM because efficient transfer by ORP5/8 requires PS in the acceptor membrane. However, as the reviewer pointed out, there is still a possibility that PTDSS1 inhibition reduces expression of ORP5/8 or disturbs their localization, thereby inhibiting PI4P transfer to the

ER. PTDSS1 inhibition did not affect the mRNA expression of endogenous ORP5 (OSBPL5) and ORP8 (OSBPL8) (**Supplementary Fig. 7A**) but induced the recruitment of ectopically expressed mCherry-ORP5 and mCherry-ORP8 to the ER-PM contact sites, rather than disturb their localization (**Supplementary Fig. 7G-7I**). Depletion of PM PI4P by treatment with PI4KIIIa inhibitor completely reversed the translocation of ORP5/8 (**Fig. 7B**), suggesting that ORP5/8 senses excess accumulation of PM PI4P to transfer it to the ER, where PI4P is dephosphorylated to PI. Nevertheless, it is less likely that PTDSS1 inhibition causes downregulation or mislocalization of ORP5/8.

2-3. Nir2/3 contributes to PI and PIP elevation in response to PTDSS1 inhibition.

Double-knockout of ORP5/8 (ORP5/8-DKO) in Ramos cells increased the PIP levels but to a lesser degree than PTDSS1i treatment (**Fig. 7C**). Moreover, we found that PTDSS1 inhibition still increased the PIP levels even in ORP5/8-DKO cells, indicating that additional mechanism involves in PIPs regulation by PTDSS1i. As the reviewer suggested, Nir2/3 potentially contributes to PI elevation in PTDSS1i-treated cells because PTDSS1i inhibition also resulted in an elevation of PA (**Fig. 1C**), which is known to stimulate the Nir2/3-mediated PA/PI countertransport. While Nir2/3-DKO did not affect PI and PIP levels in the basal condition, it significantly impaired the elevation of PI and PIP induced by PTDSS1 inhibition (**Fig. 7C, 7D, and 7G**). Nir2/3-DKO cells also showed the concomitant accumulation of PA in response to PTDSS1 inhibition (**Fig. 7G**), suggesting that PTDSS1 inhibition enhanced Nir2/3-mediated PA to PI conversion, the so-called 'PI-cycle'.

Based on those observations, we propose the revised model (**Fig. 8**), where both deceleration of PS/PI4P countertransport by ORP5/8 and acceleration of PI-cycle by Nir2/3 contribute to the elevation of phosphoinositides under PTDSS1 inhibition. We added Fig. 7B, 7C, 7D, 7G, and Supplementary Fig. 7, and accordingly changed the Results (lines 236-275) and Discussion (lines 306-331) in the revised manuscript.

3. What is the mechanism behind the elevated production of IP1 in PTDSS1 KO cells? Do the authors propose that this is due to higher levels of PIP2 in PTDSS1 KO? If so, are the localization and behavior of PLCg2 changed in PTDSS1 KO cells with/without BCR stimulation? Additionally, can the authors test if reduction of PIP2 and/or PIP3 (by

overexpression of PIP2 or PIP3 phosphatases or some other acute recruitment techniques) rescues the elevated production of IP₁ and Ca²⁺ response in PTDSS1 KO cells? If the authors propose different mechanism(s), please provide such evidence.

Response 3: We propose that PTDSS1 inhibition primarily affects PIP levels as mentioned in **Response 2-1**. Following the reviewer's suggestion, we examined the effects of acute PI4P depletion by treatment with PI4KIIIa inhibitor on the elevated production of IP₁ and Ca²⁺ response in PTDSS1i-treated Ramos cells. Treatment with PI4KIIIa inhibitor specifically reduced PM PI4P without affecting PM PI(4,5)P₂ (**Fig. 6D**). We found that the enhanced IP₁ production and Ca²⁺ response in PTDSS1i-treated cells were significantly attenuated by treatment with PI4KIIIa inhibitor (**Fig. 6E and 6F**). We confirmed that PTDSS1 inhibition did not affect the phosphorylation status of PLC γ 2, as well as subcellular localization (**Supplementary Fig. 5B and 5C**). Therefore, we concluded that PTDSS1 inhibition primarily increases the PM PI4P level, thereby promoting the flow of downstream BCR-derived IP₁ production. We added Fig. 6D, 6E, 6F, and Supplementary Fig. 5B and 5C, and accordingly changed the Results (lines 219-227) in the revised manuscript.

4. Are organelles such as mitochondria, endosomes, lysosomes, Golgi and the ER, and the PM in PTDSS1 KO and PTDSS1i-treated cells not overtly affected in their number, position and morphology with/without BCR stimulation?

Response 4: Following the reviewer's suggestion, we examined the effects of PTDSS1 inhibition on intracellular organelles by immunocytochemical staining for organelle-resident proteins. We did not observe overt abnormalities in Golgi, mitochondria, endosomes, or plasma membranes (**Supplementary Fig. 3**), even in the presence of BCR stimulation. Interestingly, while PTDSS1 inhibition did not affect the morphology of the perinuclear ER as judged by staining with an anti-KDEL antibody (**Supplementary Fig. 3**), it increased the ER-PM contact sites probably through PI4P accumulation in the PM (**Fig. 7B and Supplementary Fig. 7I**). We added Supplementary Fig. 3, and accordingly changed the Results (lines 145-146) and Discussion (lines 322-327) in the revised manuscript.

Minor points:

1. As a control for rescue experiments by adding PS in the culture media, irrelevant lipids such as PC should be tested.

Response 5: Following the reviewer's suggestion, we tested the rescue effect of PS with PC as an irrelevant lipid control. We confirmed that C36:2-PS, but not C36:2-PC, rescued the elevation of BCR-induced Ca^{2+} response and induced apoptosis in PTDSS1-KO Ramos cells (**Fig. 4B and 4C**). We added Fig. 4B and 4C, and accordingly changed the Results (line 166) in the revised manuscript.

2. Please indicate the statistical values for Fig. 5G, 6D and 6E. Are they statistically significant?

Response 6: Following the reviewer's suggestion, we indicate the statistical P values (**Fig. 5G** in the revised manuscript).

Reviewer #2 (Comments to the Authors (Required)):

The manuscript reports an interesting effect of PTDSS1 targeting on the survival of lymphoma cells. Instead of synthetic lethality with PTDSS2, the lymphoma cells are sensitive in part through enhanced BCR-mediated apoptosis. This indicates that PS synthesis and balanced levels of PI and PE are important selectively in these haematopoietic cancers. The results are novel and I find the experiments interesting, informative and mostly well done. The conclusions about how PS synthesis regulates oncogenic (and tumour-suppressive) BCR signalling should be important for the field. I do have some questions, however, about the mechanisms by which the phospholipid imbalance causes apoptosis.

Response: We would like to thank the reviewer #2 for reviewing our manuscript and offering constructive suggestions that help strengthen our conclusion. We have addressed all of the reviewer's concerns in the point-by-point responses as follows, especially with the clear evidence for the involvement of enhanced Ca^{2+} signaling in PTDSS1i-mediated cell death as described in the response to the major issue #1

Major issues:

1. The experiments on BCR-induced apoptosis are very interesting and so is the idea that enhanced BCR calcium signalling is responsible for the apoptosis of the PTDSS1i-treated cells. I don't think any previous studies actually showed spontaneous calcium oscillations in lymphoma studies, so this is novel. Implicating calcium signalling in apoptosis is reasonable (see e.g. Akkaya Nat Immune 2018), however, there are no experiments that directly support this conclusion (as opposed to another signalling pathway downstream of the BCR causing the apoptosis). Can the authors selectively block calcium signalling (e.g. inhibition or KO of Orai etc.) to see if this rescues the apoptosis mediated by PTDSS1i?

Response 1: Following the reviewer's suggestion, we examined whether pharmacological inhibition of Ca^{2+} signaling rescues the PTDSS1i-induced apoptosis using 2-

aminoethoxydiphenyl borate (2-APB), which is widely used to inhibit both IP₃ receptor-mediated Ca²⁺ release and Orai1-mediated Ca²⁺ entry. We found that 2-APB significantly reduced the apoptosis induced by PTDSS1 inhibition (**Fig. 5H**). This finding substantially strengthens the proposed model for the mechanism of action of PTDSS1i to B cell lymphoma. We added Fig. 5H, and accordingly changed the Results (lines 195-198) in the revised manuscript.

2. The rescue experiments, which knock out BCR signalling proteins or use IgM-negative Ramos cells are critical, but need some clarification. First, chronic oncogenic BCR signalling has been reserved for ABC-type DLBCL and is associated with BTK-PLCγ2-NFκB signalling. In Burkitt lymphoma cells (Ramos) and GCB-type DLBCL (SUDHL6), the oncogenic effects are driven by the "tonic" CD19-PI3K pathway. The cited papers actually refer to a range of leukaemia and lymphoma studies that describe a variety of mechanisms. So it would be clearer not to call the BCR signalling "chronic". Second, it is expected that KO of CD19 or CD79B will retard the growth of the Ramos cells. Thus, the experiment in Fig 5G is problematic as the data has been normalised, making it impossible to see this. Without taking this baseline effect into account, it is difficult to judge the rescue effect on the PTDSS1i-treated cells. The KO of PLCγ2 is a better experiment, because PLCγ2 is not expected to be essential for the growth of Ramos cells, but does regulate calcium signalling. Unfortunately, the PLCγ2 KO cells have been obtained by selection involving resistance to apoptosis, making it unclear whether it is PLCγ2 or the selection that is responsible for the effect. Ideally, PLCγ2 KO could be done with an expression marker that could allow monitoring the growth of the KO cells compared to WT without this selection. Similarly, the experiment with the Ramos cells that spontaneously lost IgM is problematic, because we do not know how these cells adapted to their BCR loss.

Finally, I understand from the discussion that the authors are aware that the rescue here is not complete and thus other BCR-independent mechanisms are at play (PE?). But it may be fair to bring this point here as well.

Response 2: Following the reviewer's suggestion, we changed the word "chronic BCR

signaling" to simply "BCR" (line 178) in the revised manuscript. As the reviewer pointed out, KO of CD19 or CD79B itself potentially retards the growth of Ramos cells, due to the loss of tonic BCR signaling. However, we did not observe any growth retardation in these KO cells compared to the parent cells, as judged by the absolute cell concentrations (million cells per mL) in the absence of PTDSS1i: 1.08 ± 0.07 for the parent, 0.99 ± 0.11 for sgCD22, 1.07 ± 0.11 for sgCD19, and 1.06 ± 0.23 for sgCD79B (**Fig. 5G**). This observation is consistent with a previous report demonstrating that BCR-KO does not affect the growth curve of Ramos cells under non-competitive conditions (*EMBO J.*, 2018, PMID: 29669863). Based on those findings, the baseline effect is less likely to account for the rescue effects of KO in **Fig. 5G**. For clarity on this point, we specified the absolute cell concentrations in the figure legend of **Fig. 5G** (lines 940-943). Following the reviewer's comment, we now understand the problematic bias in the use of the selected PLCg2-KO or naturally occurring BCR-negative cells and decided not to present the data (sgPLCg2 in Fig. 5G and Fig. 5H in the original manuscript) in the revised manuscript. We tried the stable KD of PLCg2 with an expression marker EGFP based on a lentiviral system, however, the lentiviral infection itself substantially attenuated the cytotoxic sensitivity to PTDSS1i irrespective of shRNA sequences, making it difficult to judge the effects of KD. Therefore, the data presented in the revised manuscript could not directly prove the involvement of spontaneous PLCg2 activation. Nevertheless, we believe that those findings presented in **Fig. 5G** can still support the cytotoxic role of aberrant BCR signaling through CD79B under PTDSS1 inhibition and the currently proposed model (**Fig. 8**).

As the reviewer pointed out, not only PIP elevation but also PE reduction contributes to the cytotoxicity of PTDSS1 inhibition (**Supplementary Fig. 9**). Following the reviewer's suggestion, we refer to the findings on PE in the Results (lines 277-285) following the findings on PIPs regulation in the revised manuscript.

3. The authors do not take into account PIP3 levels and phosphorylation of Akt. This is a critical mechanisms required for the growth of Ramos cells. Are PIP3 levels and Akt phosphorylation affected by PTDSS1 inhibition or knock out?

Response 3: Following the reviewer's suggestion, we examined the effects of PTDSS1 inhibition on the levels of PI(3,4,5)P₃ and phosphorylation of Akt. In contrast to PI4P (**Fig. 6C**), we found a lower PM/Cyto ratio for PI(3,4,5)P₃ probe in PTDSS1i-treated cells, even under the BCR ligation condition that significantly increased the PM/Cyto ratio of PI(3,4,5)P₃ probe (**Supplementary Fig. 6F**). This observation was consistent with the downregulation of CD19 (**Supplementary Fig. 6A and 6G**), which promotes PI(3,4,5)P₃ production via PI3K activation, although the mechanistic link between PTDSS1 and CD19 still remains unknown. Nevertheless, PTDSS1 inhibition did not significantly affect the phosphorylation status of Akt in basal or BCR-stimulated conditions (**Supplementary Fig. 6G**), as judged by phospho-Akt flowcytometry (*Blood*, 2017, PMID: 28011673). Therefore, we concluded that PI(3,4,5)P₃-Akt pathway did not substantially contribute to the cytotoxicity of PTDSS1 inhibition. We added Fig. 6C and Supplementary Fig. 6F and 6G, and accordingly changed the Results (lines 210-212 and 228-234) in the revised manuscript.

Minor comments:

4. PTDSS1 WB shows multiple bands. The authors may want to illustrate their confidence that this is the correct band and highlight it better.

Response 4: Following the reviewer's suggestion, we specified the correct bands and the non-specific bands appearing above the correct band in both the Figure and Figure Legends (lines 1025-1026) in the revised manuscript.

5. Line 124 is unclear. The A549 cells are not affected, but the SUDHL6 are.

Response 5: Following the reviewer's suggestion, we clarified the sentence to "PTDSS1-KO SU-DHL-6 lymphoma, but not A549 lung carcinoma, also showed significant cell death." (lines 122-123) in the revised manuscript.

6. Fig 3b the red line for the PTSS1 KO is invisible, (it should overlap with 0?).

Response 6: Following the reviewer's comment, we specified that the curve for PTDSS1-KO overlaps with the x -axis in the Figure Legends (line 899) in the revised manuscript.

7. Fig 5 - the numbers of spikes should be converted to frequency so that they are independent of the duration of data collection.

Response 7: Following the reviewer's suggestion, we converted the y-axis to the frequency of Ca^{2+} spikes (spikes per sec) in Fig. 5A and 5C, and the Figure Legends (line 934) in the revised manuscript.

8. Why is the KO of ORP5+8 increasing the total levels of PIPs? According to Fig 7, this will affect translocation. But as far as I can see the lipid measurements are from the whole cell, not just the PM. Does the translocation, lead to destruction? Please explain.

Response 8: Yes. The translocation of PI4P to the ER by ORP5/8 leads to dephosphorylation of PI4P because the PI4P phosphatase (SACM1L) is abundantly expressed and localized in the ER (**Supplementary Fig. 7F**). Following the reviewer's comment, we explained this machinery in the Result (lines 243-245) and in the schematic figure (**Fig. 8**) in the revised manuscript.

9. I suggest that supplementary Fig 5C,D is described in "Results" to help to explain that lymphoma also likely suffer from reduced PE. Since the focus is on the PIP2-IP3-Calcium signalling, this is fine to leave open, but it is important to acknowledge in the main part of the paper.

Response 8: As the reviewer pointed out, not only elevated PIP but also reduced PE is responsible for the PTDSS1i-induced cytotoxicity. Following the reviewer's suggestion, we refer to the findings on PE in the Results (lines 277-285) in the revised manuscript.

Reviewer #3 (Comments to the Authors (Required)):

In this manuscript by Omi et al., the authors showed that inhibition or depletion of PTDSS1, a key enzyme for phosphatidylserine (PS) synthesis, results in increased phosphatidylinositol (PI) lipids, cytosolic calcium signals, and cell death in B cell receptor (BCR)-positive lymphoma cell lines. The authors suggest that targeting PS synthesis may be a new strategy to treat B cell lymphoma.

Response: We would like to thank the reviewer #3 for reviewing our manuscript and offering constructive suggestions that help strengthen our conclusion. We have addressed all of the reviewer's concerns in the point-by-point responses as follows, especially with more mechanistic insights for the regulation of PI and PIPs by PTDSS1i as described in the response to the first and second comments.

This manuscript presents a lot of results. Nevertheless, there is a lack of new mechanistic insights into the elevated PI lipids and calcium signaling caused by PTDSS1 deficiency. The authors proposed a model that aberrant PIP/calcium levels and apoptosis are caused by abolished PS/PIP exchange at ER-PM contact sites. However, the effects of PTDSS1 deficiency on PI lipids are much bigger than knockdown of both ORP5 and ORP8 that mediate PS/PIP exchange at ER-PM contact sites (note: the differences in the scale of the Y-axis of Figures 6B/6C vs Figures 6F/6G). Also, it's unclear whether ORP5/ORP8 expression is significantly reduced in ORP5/ORP8 double knockdown Ramos Cells in these experiments. The results do not provide sufficient support for the proposed model.

Response 1: As the reviewer points out, ORP5/8-double knockdown only partially phenocopy the effects of PTDSS1 inhibition. This might be due to the possibility that shRNAs could not efficiently reduce the protein levels of ORP5/8 which we have not defined in the original manuscript, or that the additional mechanism(s) other than ORP5/8 contribute to the PIPs elevation in PTDSS1i-treated cells. To avoid the ambiguity of the knockdown experiment, we newly generated ORP5/8-double KO (ORP5/8-DKO) Ramos cells by

CRISPR-Cas9 (**Supplementary Fig. 8A-B**) and analyzed their phenotypes (elevation of PIP), comparing to the phenotypes of PTDSSi-treated cells. ORP5/8-DKO cells showed a significant elevation of PIP levels compared to parental cells (**Fig. 7C-E**). However, we found again that the ORP5/8-DKO only partially phenocopied the effects of PTDSS1 inhibition and that PTDSS1 inhibition still increased PIP levels even in ORP5/8-DKO cells. These observations suggested that an additional mechanism other than ORP5/8 contributes to PIP elevation under PTDSS1 inhibition. As the reviewer pointed out in the second comment below, Nir2/3 is a possible candidate for such additional machinery because PTDSS1i increased not only PIP but also PA which has been shown to stimulate Nir2/3-mediated 'PI-cycle'. We, therefore, performed additional experiments to define the involvement of Nir2/3 in PTDSS1i-mediated PI/PIP elevation and address the reviewer's comment below (**Response 2**).

It is interesting to note that PA is elevated, like PI lipids, in most PTDSS1i-treated cancer cells (Figure 1). Elevated PA has been shown to trigger targeting of PI/PA transfer proteins Nir2/Nir3 to ER-PM contact sites for replenishing plasma membrane PI(4,5)P2 following receptor-induced PLC activation (PMID: 25887399). Nevertheless, the elevation of PA in PTDSS1 inhibited cells was not discussed in this study.

Response 2: Following the reviewer's comment, we generated Nir2/3-DKO Ramos cells (**Supplementary Fig. 8C-D**) and analyzed their PIP content, comparing to the PTDSS1i-treated parent cells. In contrast to ORP5/8, Nir2/3-DKO significantly impaired the elevation of both PI and PIP levels, as well as BCR-derived Ca²⁺ response, under PTDSS1 inhibition (**Fig. 7C-E**). We found the further accumulation of PA in Nir2/3-DKO cells in response to PTDSS1 inhibition (**Fig. 7G**). Based on these observations, we revised our working model in which both a deceleration of PI4P-PS countertransport by ORP5/8 and an acceleration of the PI-cycle by Nir2/3 contribute to the increased PIP and BCR hyperactivation under PTDSS1 inhibition (**Fig. 8**). We added Fig. 7 and Supplementary Fig. 8, and accordingly changed the Results (lines 240-275) and Discussion (lines 306-331) in the revised manuscript.

Hyper-responsiveness to T cell receptor signaling has been shown in PTEN-deficient Jurkat T cells (PMID: 10958690). As shown in Figure 1, Jurkat cells exhibited higher PI lipids and cell death, like B lymphoma cell lines used in most figures of this manuscript, following PTDSS1 inhibitor treatment. Is elevated BCR signaling caused by PTEN deficiency in B lymphoma lines used in this study? If so, PTEN should be incorporated into the proposed model of this manuscript.

Response 3: As the reviewer pointed out, PTEN deficiency observed in Jurkat cells potentially contributes to the hyper-responsiveness of BCR signaling in B cell lymphoma under PTDSS1 inhibition. However, in contrast to Jurkat cells, previous studies reported the protein expression of PTEN in Ramos (PMID: 31719683), SU-DHL-6 (PMID: 9787181), SU-DHL-2 (PMID: 12808067), and Jeko-1 (PMID: 18339899). Moreover, we did not observe the increase in PM PI(3,4,5)P₃ (**Supplementary Fig. 6F**) or enhanced phosphorylation of Akt, a downstream effector of PI(3,4,5)P₃, in PTDSS1i-treated Ramos cells (**Supplementary Fig. 6G**). Therefore, PTEN deficiency is less likely to be involved in the proposed model for B cell lymphoma used in our study.

October 27, 2023

RE: JCB Manuscript #202212074R-A

Prof. Junken Aoki
The University of Tokyo
Department of Health Chemistry, Graduate School of Pharmaceutical Sciences
7-3-1, Hongo
Bunkyo-ku, Tokyo 113-0033
Japan

Dear Prof. Aoki:

Thank you for submitting your revised manuscript entitled "Phosphatidylserine Synthesis Controls Oncogenic B Cell Receptor Signaling in B Cell Lymphoma". We would be happy to publish your paper in JCB pending the remaining requests for text changes sought by Reviewers 2 and 3, and final revisions necessary to meet our formatting guidelines (see details below).

A. MANUSCRIPT ORGANIZATION AND FORMATTING:

Full guidelines are available on our Instructions for Authors page, <http://jcb.rupress.org/submission-guidelines#revised>. Submission of a paper that does not conform to JCB guidelines will delay the acceptance of your manuscript.

1) Text limits: Character count for Articles is < 40,000, not including spaces. Count includes abstract, introduction, results, discussion, and acknowledgments. Count does not include title page, figure legends, materials and methods, references, tables, or supplemental legends.

2) Figures limits: Articles may have up to 10 main figures and 5 supplemental figures/tables.

** Please lower the number of supplemental figures. This can be accomplished by making supplemental figures into main figures, merging supplemental figures, and adding supplemental figure panels to main figures. Please also note that figures cannot span more than one page.

3) Figure formatting: Scale bars must be present on all microscopy images, including inset magnifications. Molecular weight or nucleic acid size markers must be included on all gel electrophoresis. Please avoid pairing red and green for images and graphs to ensure legibility for color-blind readers. If red and green are paired for images, please ensure that the particular red and green hues used in micrographs are distinctive with any of the colorblind types. If not, please modify colors accordingly or provide separate images of the individual channels.

** Please include molecular weight markers for blots in Figure S2 and Figure S8.

4) Statistical analysis: Error bars on graphic representations of numerical data must be clearly described in the figure legend. The number of independent data points (n) represented in a graph must be indicated in the legend. Statistical methods should be explained in full in the materials and methods. For figures presenting pooled data the statistical measure should be defined in the figure legends. Please also be sure to indicate the statistical tests used in each of your experiments (either in the figure legend itself or in a separate methods section) as well as the parameters of the test (for example, if you ran a t-test, please indicate if it was one- or two-sided, etc.). Also, if you used parametric tests, please indicate if the data distribution was tested for normality (and if so, how). If not, you must state something to the effect that "Data distribution was assumed to be normal but this was not formally tested."

5) Abstract and title: The abstract should be no longer than 160 words and should communicate the significance of the paper for a general audience. The title should be less than 100 characters including spaces. Make the title concise but accessible to a general readership.

6) Materials and methods: Should be comprehensive and not simply reference a previous publication for details on how an experiment was performed. Please provide full descriptions in the text for readers who may not have access to referenced manuscripts. We also provide a report from SciScore and an associate score, which we encourage you to use as a means of evaluating and improving the methods section.

7) Please be sure to provide the sequences for all of your primers/oligos and RNAi constructs in the materials and methods. You must also indicate in the methods the source, species, and catalog numbers (where appropriate) for all of your antibodies. Please also indicate the acquisition and quantification methods for immunoblotting/western blots.

8) Microscope image acquisition: The following information must be provided about the acquisition and processing of images:

- Make and model of microscope
- Type, magnification, and numerical aperture of the objective lenses
- Temperature
- Imaging medium
- Fluorochromes
- Camera make and model
- Acquisition software
- Any software used for image processing subsequent to data acquisition. Please include details and types of operations involved (e.g., type of deconvolution, 3D reconstitutions, surface or volume rendering, gamma adjustments, etc.).

** Please include the above details in the methods section on microscopy.

10) Supplemental materials: There are strict limits on the allowable amount of supplemental data. Articles may have up to 5 supplemental figures. Please also note that tables, like figures, should be provided as individual, editable files. A summary of all supplemental material should appear at the end of the Materials and methods section.

13) ORCID IDs: ORCID IDs are unique identifiers allowing researchers to create a record of their various scholarly contributions in a single place. At resubmission of your final files, please include an ORCID ID for all authors.

Please note that JCB now requires authors to submit Source Data used to generate figures containing gels and Western blots with all revised manuscripts. This Source Data consists of fully uncropped and unprocessed images for each gel/blot displayed in the main and supplemental figures. Since your paper includes cropped gel and/or blot images, please be sure to provide one Source Data file for each figure that contains gels and/or blots along with your revised manuscript files. File names for Source Data figures should be alphanumeric without any spaces or special characters (i.e., SourceDataF#, where F# refers to the associated main figure number or SourceDataFS# for those associated with Supplementary figures). The lanes of the gels/blots should be labeled as they are in the associated figure, the place where cropping was applied should be marked (with a box), and molecular weight/size standards should be labeled wherever possible. Source Data files will be directly linked to specific figures in the published article.

Journal of Cell Biology now requires a data availability statement for all research article submissions. These statements will be published in the article directly above the Acknowledgments. The statement should address all data underlying the research presented in the manuscript. Please visit the JCB instructions for authors for guidelines and examples of statements at (<https://rupress.org/jcb/pages/editorial-policies#data-availability-statement>).

WHEN APPROPRIATE: The source code for all custom computational methods published in JCB must be made freely available as supplemental material hosted at www.jcb.org. Please contact the JCB Editorial Office to find out how to submit your custom macros, code for custom algorithms, etc. Generally, these are provided as raw code in a .txt file or as other file types in a .zip file. Please also include a one-sentence summary of each file in the Online Supplemental Material paragraph of your manuscript.

B. FINAL FILES:

Thank you for this interesting contribution, we look forward to publishing your paper in Journal of Cell Biology.

Sincerely,

William Prinz
Monitoring Editor
Journal of Cell Biology

Tim Fessenden
Scientific Editor
Journal of Cell Biology

Reviewer #1 (Comments to the Authors (Required)):

The authors have addressed all concerns and now I support the publication of the revised version of the manuscript.

Reviewer #2 (Comments to the Authors (Required)):

The authors addressed my previous comments and I think the paper is a nice addition to the understanding of B lymphoma biology and BCR signalling.

I just suggest adding a sentence to line 190 explaining that KO of BCR components or CD19 did not affect growth of the cells in these non-competitive cultures, although the cells are expected to lose fitness (PMID: 29669863).

Reviewer #3 (Comments to the Authors (Required)):

In this revised manuscript by Omi et al, the authors have included a lot of new data and discussions to address my comments on the mechanisms linking PS metabolism with the PI cycle/calcium signaling. My previous concerns have been sufficiently

addressed. The manuscript is nicely written. Nevertheless, I suggest the following minor edits for clarity and accuracy.

Line 248: cite the reference for GFP-MAPPER (Chang et al, 2013 Cell Reports, PMID: 24183667).

Line 264: include the reference (Chang and Liou, 2015 JBC, PMID: 25887399) which showed that Nir2 and Nir3 mediate PI transfer from the ER to the PM for the PI cycle. Kim et al 2015 only investigated Nir2.

Line 323-324: "...it increased the PM-ER contact sites probably through PI4P accumulation in the PM..." Should be changed to "it increased the recruitment of ORP8 and MAPPER to PM-ER contact sites through PI4P and PI(4,5)P2 accumulation in the PM..." to reflect the observed results. Localization of MAPPER to PM-ER contact sites is dependent on PM PI(4,5)P2 (Chang et al. 2013 Cell Reports PMID: 24183667). Since PTDSS1i treatment increased PM PI(4,5)P2 (Fig. S6 B), it can increase MAPPER signal at the PM (Fig. S7 I) without actually increasing PM-ER contact sites. For the same reason, the following sentence "The increased PM-ER contact sites....." (Line 324-327) should be changed accordingly or deleted.